



# Representation of soil hydrology in permafrost regions may explain large part of inter-model spread in simulated Arctic and subarctic climate

Philipp de Vrese[1], Goran Georgievski[1], Jesus Fidel Gonzalez Rouco[2], Dirk Notz[1,3], Tobias Stacke[1], Norman Julius Steinert[4], Stiig Wilkenskjeld[1], and Victor Brovkin[1,5]

[1]Max Planck Institute for Meteorology, The Ocean in the Earth System, Hamburg, 20146, Germany
[2]University Complutense of Madrid, Department of Earth Physics and Astrophysics, Madrid, 28040, Spain
[3]University of Hamburg, Faculty of Mathematics, Informatics and Natural Sciences, Hamburg, 20146, Germany
[4]Norwegian Research Centre Climate and Environment, Bjerknes Centre for Climate Research, Bergen, 5007, Norway
[5]University of Hamburg, Center for Earth System Research and Sustainability, Hamburg, 20146, Germany

**Correspondence:** Philipp de Vrese (philipp.de-vrese@mpimet.mpg.de)

**Abstract.** The current generation of Earth system models exhibits large inter-model differences in the simulated climate of the Arctic and subarctic zone, with differences in model structure and parametrizations being one of the main sources of uncertainty. One particularly challenging aspect in modelling is the representation of terrestrial processes in permafrost-affected regions, which are often governed by spatial heterogeneity far below the resolution of the models' land surface components.

Here, we use the MPI Earth System model to investigate how different plausible assumptions for the representation of the permafrost hydrology modulate the land-atmosphere interactions and how the resulting feedbacks affect not only the regional and global climate, but also our ability to predict whether the high latitudes will become wetter or drier in a warmer future. Focusing on two idealized setups that induce comparatively "wet" or "dry" conditions in regions that are presently affected by permafrost, we find that the parameter settings determine the direction of the 21st-century trend in the simulated soil water

content and result in substantial differences in the land-atmosphere exchange of energy and moisture. The latter leads to differences in the simulated cloud cover and thus in the planetary energy uptake. The respective effects are so pronounced that uncertainties in the representation of the Arctic hydrological cycle can help to explain a large fraction of the inter-model spread in regional surface temperatures and precipitation. Furthermore, they affect a range of components of the Earth system as far to the south as the tropics. With both setups being similarly plausible, our findings highlight the need for more observational

constraints on the permafrost hydrology to reduce the inter-model spread in Arctic climate projections.



# 1 Introduction

Earth System Models (ESMs) are our primary tool for projecting the coupled dynamics of the climate and biogeochemistry under future emission scenarios (Flato, 2011; Stocker et al., 2013), with the ensemble of simulations from the Coupled Model Intercomparison Project (CMIP; Taylor et al., 2012; Eyring et al., 2016)) providing an important basis for policy making (IPCC, 2021). But while ESMs agree on the general relation between increasing greenhouse gas concentrations and rising temperatures, there are substantial differences between the climate states and trajectories simulated by individual models. These inter-model differences are particularly prominent in the northern high latitudes, where ESMs estimate very different sea ice concentrations (Notz and SIMIP Community, 2020; Davy and Outten, 2020) as well as different surface temperatures, evapotranspiration and precipitation rates (Fig. 1). The region is of special interest not only because polar amplification causes temperatures there to increase at least twice as fast as the global average (Brown and Romanovsky, 2008; Stocker et al., 2013; Biskaborn et al., 2019), but also because Arctic and subarctic soils contain roughly 1100 - 1700 Gt of carbon, which is about twice as much as the Earth's atmosphere (Zimov et al., 2006; Schirrmeister et al., 2011; Strauss et al., 2015; Bernal et al., 2016; Vasilchuk and Vasilchuk, 2017; Tarnocai et al., 2009; Hugelius et al., 2013, 2014). Currently, the majority of these soil carbon pools is effectively inert as it is contained within permafrost — perennially frozen ground — but more and more of the organic matter will become vulnerable to decomposition as a consequence of global warming. The resulting $CO_2$ and $CH_4$ emissions further increase the rise in temperatures, making the permafrost carbon feedback an important but highly uncertain terrestrial climate feedback (Zimov et al., 2006; Schaefer et al., 2014; MacDougall et al., 2015; Schuur et al., 2015; Comyn-Platt et al., 2018; Gasser et al., 2018; Lenton et al., 2019; Randers and Goluke, 2020; Turetsky et al., 2020; de Vrese et al., 2021; Natali et al., 2021).

Studies have identified differences in model structure and parametrizations as one of the main sources of uncertainty in Arctic climate change projections (Hodson et al., 2012; Lehner et al., 2020; Bonan et al., 2021), but it is difficult to attribute the inter-model climate variability to differences in specific model components. With respect to the land surface, it appears likely that the treatment of snow is a main contributor to the model uncertainty. The high-latitude snow cover lasts for the majority of the year and differences in the simulated snowpack have been shown to often coincide with large differences in surface and subsurface temperatures (Paquin and Sushama, 2014; Ekici et al., 2015; Melo-Aguilar et al., 2018; Mudryk et al., 2020; Menard et al., 2021). During the snow-free season, the land-atmosphere exchange of energy, moisture and momentum is determined by a number of soil and vegetation properties, all of which depend — directly or indirectly — on the representation of the terrestrial hydrology (Seneviratne et al., 2010). The partitioning of the net radiation into latent and sensible heat flux, which is a key factor in the development of the near-surface temperatures, cloud formation and precipitation, depend on the amount of water that can be evaporated and transpired. The albedo of bare ground is influenced by the soil wetness at the surface, while the extent of these bare areas is partly determined by the root-zone soil moisture as an important factor for the vegetation cover. The latter also determines the surface albedo in vegetated areas and affects the exchange of momentum via its effect on the surface roughness. Thus, with almost every aspect of the land-atmosphere interactions being affected by the availability





of liquid water, it is plausible that the representation of the soil hydrology in numerical models is a major contributor to the inter-model climate variability.

Representing the soil hydrology of the Arctic and subarctic zone with coarse resolution land surface models (LSM) is especially challenging because the hydrology is often determined by small-scale landscape heterogeneity and affected by processes that are spatially and temporarily very confined. Prominent examples are the spatial soil moisture variability of the polygonal tundra (Cresto Aleina et al., 2013) and geomorphological processes linked to soil ice, including thermokarst features, thaw lake dynamics and ground subsidence (Jorgenson et al., 2006; O'Donnell et al., 2011; Liljedahl et al., 2016; Serreze et al.,

2000; Jafarov et al., 2018; Nitzbon et al., 2019; Andresen and Lougheed, 2015). Permafrost plays a key role in the terrestrial hydrological cycle because the presence of ice modulates the thermophysical soil properties as well as infiltration rates and the vertical and lateral movement of water through the ground. And while LSMs will never be able to capture the full multitude of effects connected to surface and subsurface heterogeneity, the level of realism in the representation of physical and biogeochemical permafrost-related processes and effects has increased substantially over the past years (McGuire et al.,

2016; Chadburn et al., 2017; Fisher and Koven, 2020; Blyth et al., 2021). Amongst others, many models now account for the inhibition of vertical soil moisture fluxes, which often leads to the formation of a saturated zone above the permafrost table, or the thermal insulation of the soil due to organic matter (Painter et al., 2012; Swenson et al., 2012; Toride et al., 2013; Walvoord and Kurylyk, 2016).

These model developments help improve our understanding of the ways permafrost affects the regional climate and the terrestrial carbon cycle. Still, it remains unclear how differences in the treatment of the soil hydrology in permafrost regions relate to the inter-model climate variability in the present generation of ESMs. In parts, this is because it is next to impossible to determine whether differences in Arctic and subarctic climate have local or non-local causes when comparing simulations with different fully-coupled models. To isolate local effects, most LSMs can be run in a standalone-mode, forced with prescribed

atmospheric states, precipitation rates and radiative fluxes. A recent model intercomparison using such setup showed that in permafrost regions, commonly used LSMs exhibit vastly different hydrological responses to similar atmospheric conditions (Andresen et al., 2020). Neither do the models agree on the magnitude of historical and present-day hydrological states and fluxes, nor on the question whether the high latitudes will become wetter or drier when the permafrost retreats in the future, or at least whether or not the soils contain more water (Fig. 2). Such comparisons strongly suggest that the representation of

terrestrial processes is highly relevant for the simulated climate in permafrost regions, but they do not allow one to infer the extent to which differences in the hydrology schemes contribute to the inter-model spread of the Arctic and subarctic climate. On one hand, the differences between the LSMs are not limited to the soil hydrology but extend to thermophysical processes, vegetation dynamics and the coupling to the atmosphere. On the other hand, all land-atmosphere feedbacks are omitted in the standalone-mode and it is not clear whether these feedbacks would amplify or decrease the inter-model differences when the

LSMs are coupled to an atmospheric model.





In this study, we use an adapted version of the Max-Planck-Institute for Meteorology's Earth system model MPI-ESM to estimate how uncertainties in the parametrization of the permafrost hydrology translate into uncertainty in simulated climate. The modifications to the MPI-ESM allow us to compare simulations in which the representation of terrestrial processes in permafrost-affected grid cells differs, while all other processes in the land, the atmosphere and the ocean components are represented identically. In this way, we can ensure that all differences in the simulated climate can be traced back to differences in the soil hydrology scheme, which induce comparatively "wet" or "dry" conditions in regions that are presently affected by permafrost, while fully accounting for the resulting land-atmosphere feedbacks. Section 2 details the changes to the model and the setup of the simulations, while section 3 discusses the pathways by which the uncertainty in the soil hydrology affects the Arctic and subarctic climate and investigates the relevance of the climate effects relative to the spread of the current CMIP model ensemble. Finally, section 4 discusses the findings in a global context.

## 2 Methods

### 2.1 Model

The present investigation uses MPI-ESM with the standard versions of the atmospheric component ECHAM6 and the ocean model MPIOM (Mauritsen et al., 2019) — more specifically the versions that are used in the sixth phase of the CMIP experiments (CMIP6; Eyring et al., 2016) — and the changes to the model are limited to the land component JSBACH (version 3.2; Reick et al., 2021). In JSBACH, we implemented a mask that made it possible to execute the modified code only in those grid cells in the Arctic and subarctic zone that — at present — are affected by permafrost (Fig. 1d), while the standard model code is run in the rest of the world. The mask is not based on the permafrost extent as simulated by JSBACH — as this differs between setups — but on the observed present-day extent using data from the Northern Circumpolar Soil Carbon Database (Hugelius et al., 2014). Note that throughout the manuscript, we refer to this region as the permafrost region, even though large parts of the respective areas do not feature permafrost for the entire duration of our simulations. The changes mainly pertain to the soil hydrology scheme, including the implementation of a module that simulates the dynamics of inundated areas due to ponding at the surface, but there are also important alterations to the parametrization of thermophysical processes. The representation of the soil physics in permafrost-affected regions is largely based on the implementation of Ekici et al. (2014), who introduced a 5-layer snow scheme, the phase change of water within the soil and the effect of water on the soil's thermal properties. However, there are important differences between the implementation of Ekici et al. and the model used in the present study. These are described in more detail below.

### 2.1.1 Effect of organic matter

The extremely long carbon turnover times in the northern high latitudes result in high organic matter concentrations at the surface in large parts of the permafrost-affected regions (Carvalhais et al., 2014; Luo et al., 2019). The standard CMIP6 model does not account for the effect of this organic matter, while the version of Ekici et al. includes it in the form of a pervasive layer





located on top of the soil. This layer acts as an insulation that modifies the thermal fluxes into and out of the ground but has
no influence on the simulated soil hydrology. In contrast, the present model version assumes the properties of organic matter
in the uppermost layer of the soil column whenever the vegetation cover indicates the presence of an organic top soil layer.
For the present study, we assumed this to be the case whenever the combination of forest and grass cover exceeds a third of
the grid box area. Furthermore, our version does not limit the effects of the organic layer to the thermal processes but includes
their influence on the hydrological soil properties.

The effect on the hydrological soil properties is particularly relevant for the simulated infiltration rates in clay- and silt-rich
soils, as the higher hydraulic conductivity of organic matter facilitates the percolation of water from the surface into deeper
layers of the soil, allowing more water to infiltrate when precipitation rates or snow melt fluxes are high (Araya and Ghezzehei,
2019; Fatichi et al., 2020). At the same time the high hydraulic conductivity in combination with a larger pore volume leads
to a less saturated near-surface layer even though the latter contains more water when organic matter is included in the model.
In JSBACH the evaporation from bare areas scales directly with the relative saturation of the uppermost soil layer. Hence,
the inclusion of the effects of soil organic matter strongly reduces the evaporation from the non-vegetated fraction of the grid
cell, which is in agreement with previous studies documenting comparable LSM adaptations (Lawrence and Slater, 2007).
However, the evaporation parametrization does not allow to take into consideration differences in soil texture. These can lead
to large differences in evaporation effectiveness (ratio of actual and potential evaporation) at the same relative saturation, with
coarse mineral soils exhibiting a lower resistance than fine mineral soils (Lehmann et al., 2018). Thus, due to its simplicity, the
evaporation calculation can not account for all the effects that result from changing the soil properties from mineral to organic
soils, even if a parametrization of the evaporation effectiveness of organic soils could derived from observations. Additionally,
the structure of JSBACH, which uses one set of soil properties per grid cell, can not represent the spatial heterogeneity of the
organic matter distribution at the coarse resolutions that the model is designed for. In the fractions of the grid cell that are
unsuitable for vegetation there may be very little litter at the surface, while the bare spaces between individual plants or those
with a seasonal vegetation cover would feature a distinct organic layer. Thus, assuming the organic layer to be present in all
of the grid cell almost certainly overestimates its effect on bare soil evaporation. Conceptually, the organic layer represents the
detritus and litter accumulated on top of the soil, as well as the near-surface organic matter integrated in the soil matrix. This
also complicates modelling its effect on the simulated transpiration rates, as it is unclear to which extent the organic matter is
integrated in the soil. The above ground litter may not affect the plant water availability directly, while organic matter within the
soil increases the porosity and the available water capacity within the root zone. However, it is not clear whether an increase in
soil organic matter leads to an increase in available water capacity that is proportional to the increase in pore volume (Minasny
and McBratney, 2017). Without being able to include the entirety of effects related to soil texture or to explicitly resolve the
organic matter distribution horizontally and vertically, we include several options to account for its effect on transpiration and
evaporation, resulting in large differences in the simulated moisture recycling rates in the permafrost-affected grid cells (see
below).



### 2.1.2 Infiltration

In the standard JSBACH version, infiltration is only possible when the temperature of the first soil layer is at or above the melting point. However, in combination with the new 5-layer snow scheme introduced by Ekici et al., this formulation is problematic. In spring, the temperatures of the ground are necessarily below that of the overlying snowpack, whose temperature is 0°C during snowmelt. This results in the entire meltwater running off at the surface, while, in reality, a large fraction of the meltwater can be expected to infiltrate into the seoil (Zhang et al., 2010). In the present version, this condition was removed so that infiltration is exclusively controlled by the saturation of the near-surface soil layers and the topography within the grid cell. Furthermore, the ARNO model (Dümenil and Todini, 1992; Todini, 1996) that JSBACH uses to determine the infiltration rates based on the subgrid-scale orography, does not account for ponding effects. Instead, all water reaching the surface is either infiltrated or converted into surface runoff. In the present version of JSBACH, we implemented the WEtland-Extent-Dynamics (WEED) scheme, which adds an intermediate storage to the land surface that intercepts all rainfall and snow melt prior to infiltration or runoff generation (Stacke and Hagemann, 2012; de Vrese et al., 2021). Conceptually, this reservoir provides a minimum delay before water can infiltrate into the soil — allowing it to pond — as well as representing the (possible) formation, expansion and drainage of surface water bodies. The scheme accounts for evaporation from the reservoir and for direct infiltration under the resulting ponds, both of which depend on the grid cell fraction that is covered by wetlands. The inundated area, in turn, depends on the subgrid-scale orography and the maximum lag of the reservoir ($P_{lag}$) which is a fixed, globally uniform parameter. The largest fraction of the outgoing fluxes, however, is subdivided into surface runoff and soil infiltration according to JSBACH's standard infiltration scheme (Hagemann and Stacke, 2015). In the present model version the infiltration rates are much higher than in simulations with the standard JSBACH model, which is in large parts due to the consideration of the organic top soil layer. Thus, an additional factor $F_{ARNO}$ was implemented which allows reducing the flux from the surface storage to the ARNO scheme, with the residual outflow being allocated to the surface runoff directly, providing a straightforward way to scale the infiltration rates in permafrost-affected regions.

### 2.1.3 Evapotranspiration

JSBACH determines the vegetation's water stress and the resulting transpiration rates based on the degree of saturation within the root zone ($S_{rz}$). The latter is determined by dividing the liquid water content of the root zone by a fixed parameter, $W_{MAX,rz}$, which represents the maximum root zone soil moisture. The implementation of this approach, however, is not well suited for regions with perennially frozen ground if the phase change of soil water is represented by the model. In reality, the vegetation cover can adapt to the environmental conditions, limiting the root zone to those depths at which water is still liquid during the growing season. In the model, such an adaptation is not possible, as $W_{MAX,rz}$ is a fixed parameter. This can result in a constant water stress when the root zone extends into the perennially frozen fraction of the ground, even if there is sufficient liquid water available in the layers above the permafrost table. In the present version we mitigate this problem by accounting for the presence of ice and the model computes $S_{rz}$ relative to the ice-free pore space:





$$E_{tr}^{act} = E_{tr}^{pot} \cdot S_{rz}, \tag{1}$$

with

$$S_{rz} = \frac{W_{liq,rz}}{W_{MAX,rz}^*}, \tag{2}$$

and

$$W_{MAX,rz}^* = W_{MAX,rz} - W_{ice,rz}. \tag{3}$$

$E_{tr}^{act}$ is the simulated transpiration, $E_{tr}^{pot}$ the potential transpiration, $W_{liq,rz}$ the liquid water content of the root zone, $W_{ice,rz}$

the ice content of the root zone and $W_{MAX,rz}^*$ the adapted maximum root zone soil moisture. Furthermore, the present model version includes the option to increase $W_{MAX,rz}^*$ by the additional pore space of an organic top layer $\Delta\Phi_{org}$, which corresponds to the assumption that the organic matter is integrated in the root zone rather than being located on top of the soil and that the increase in available water capacity is proportional to the increase in pore volume (see "Effect of organic matter" above):

$$W_{MAX,rz}^* = W_{MAX,rz} + \Delta\Phi_{org} - W_{ice,rz} . \tag{4}$$

It should be noted that this option decreases the simulated transpiration rates because $W_{MAX,rz}^*$ is used to divide $W_{liq,rz}$ when determining $S_{rz}$ (see eqs. 2, 3).

A realistic parameterisation of bare soil evaporation for the Arctic and subarctic region is similarly difficult. Evaporation from

non-vegetated, snow-free soils is determined by the saturation of the uppermost soil layer ($S_{top}$), considering the liquid water content ($W_{liq,top}$) relative to a fixed parameter that represents the maximum water holding capacity of this layer ($W_{MAX,top}$). With a thickness of 6.5 cm, the first soil layer is comparatively thick, which can lead to the problem that evaporation is reduced substantially when there is ice in the top soil layer, despite an abundance of liquid water at the surface. Similar to the parametrization of transpiration, we mitigate this problem by reducing $W_{MAX,top}$ by the respective ice volume ($W_{ice,top}$),

when determining the $S_{top}$:





$$E_{bs}^{act} = E_{bs}^{pot} \cdot S_{top}, \tag{5}$$

with

$$S_{top} = \frac{W_{liq,top}}{W_{MAX,top}^*} \tag{6}$$

and

$$W_{MAX,top}^* = W_{MAX,top} - W_{ice,top}. \tag{7}$$

Here, $E_{bs}^{act}$ is the simulated bare soil evaporation, $E_{bs}^{pot}$ the potential bare soil evaporation and $W_{MAX,top}^*$ the adapted maximum water holding capacity of the uppermost soil layer. Finally, we included the option to increase $W_{MAX,top}^*$ by the additional pore space of an organic top layer. Accounting for this additional pore space corresponds to the assumption that the organic layer is present in all bare areas and has a high resistance to evaporation (see "Effect of organic matter" above), reducing the evaporation effectiveness as $W_{MAX,top}^*$ is used to divide $W_{liq,top}$ to determine $S_{top}$ (see eqs. 6, 7):

$$W_{MAX,top}^* = W_{MAX,top} + \Delta\Phi_{org} - W_{ice,top} . \tag{8}$$

### 2.1.4 Percolation, drainage and supercooled water

When pore water freezes, it blocks the pathways by which the remaining liquid water percolates into the deeper layers. As the standard JSBACH version does not include the phase change of water in the soil, the model does not account for the above effect on the vertical movement of water through the ground, and neither does the version of Ekici et al.. In the present version we included an ice impedance factor ($F_{perc,l}$) in the shape of a power law, which is common practice in present-day land surface models (Andresen et al., 2020). Here, we follow the approach of the Community Land Model (Swenson et al., 2012) and calculate $F_{perc,l}$ for soil layer $l$ as

$$F_{perc,l} = 10^{-6\frac{W_{ice,l}}{\Theta_{fc,l}}} , \tag{9}$$

with $\Theta_{fc,l}$ being the field capacity that constitutes the upper limit of the soil water content in JSBACH. In permafrost regions, the inclusion of $F_{perc,l}$ effectively prohibits percolation to the bedrock boundary, which can result in the formation of a highly saturated zone perched on top of the perennially frozen fraction of the ground. This, however, also depends on the way drainage and supercooled water are handled by the model (see below).





JSBACH's soil hydrology scheme includes two drainage components. The majority of water is assumed to drain from the
lowest hydrologically active layer at the border to the bedrock. However, in permafrost-affected regions the drainage via this
pathway is strongly limited due to the impedance of percolation in the presence of ice. The second component is a lateral
drainage from all hydrologically active layers. Here, it is assumed that water flows horizontally through the soil until it reaches
the river system or wider vertical channels — such as cracks, crevices and connected pathways in coarse material — that

provide an additional pathway by which the water reaches the border between soil and bedrock where it runs of as base flow.
Such vertical channels are assumed to be present in all grid cells at the coarse resolutions that JSBACH is typically run at.
The present model version includes the option to apply an ice impedance factor ($F_{drain,l}$) to the lateral drainage component,
corresponding to the assumption that in permafrost-affected regions, the wider vertical channels would also be blocked by ice.
Without an explicit treatment of the connected vertical channels or the excess ice in the ground, we approximate $F_{drain,l}$ in a

given soil layer $l$ by a function of the pore ice content relative to the field capacity. However, in contrast to $F_{perc,l}$, we do not
use the ice content and field capacity of layer $l$, but of all subjacent layers:

$$F_{drain,l} = 10^{-6\frac{W_{ice,sub}}{\Theta_{fc,sub}}} , \tag{10}$$

with

$$W_{ice,sub} = \sum_{i=l+1}^{n} W_{ice,i} \tag{11}$$

and

$$\Theta_{fc,sub} = \sum_{i=l+1}^{n} \Theta_{fc,i} . \tag{12}$$

$F_{perc,l}$ does not suppress the lateral drainage from layer $l$ completely, even if the subjacent layers are fully (ice) saturated.
However, in this case, only the water that exceeds the layer's field capacity is added to the drainage flux, accounting for the
possibility that in fully saturated unfrozen soil layers, lateral subsurface flow allows direct drainage into the river system, even

if the subjacent layers are frozen and the vertical channels are blocked by ice.

   A given fraction of the water within the soil — the supercooled water — may remain liquid when temperatures drop
below 0°C. In clay-rich soils, supercooled water constitutes up to a quarter of the total soil water content, as the absorptive
and capillary forces that soil particles exert on the surrounding water inhibit the freezing process (Niu and Yang, 2006).

Additionally, high salt concentrations lower the freezing point, and liquid brine lenses can even sustain microbial activity at
temperatures substantially below 0°C (Jansson and Taş, 2014). Including the effects of supercooling in LSMs is difficult, not
only because most models do not explicitly simulate the salt concentration within the soil, but also because it is not clear how
to treat the mobility of liquid water at sub-zero temperatures. Water that remains liquid due to the salt content may still move
through the surrounding soil-ice matrix, while the supercooled water that exists because of absorptive forces may be bound to





the soil particles. In the implementation of Ekici et al. the supercooled water behaves similarly to water at temperatures above
       $0°C$ — that is, it percolates through and drains from the soil. In the present version, we included the option to "immobilize"
       the supercooled water, prohibiting all fluxes below the freezing point.

## 2.2  Setups

The present investigation is mainly based on two model setups that lead to different degrees of "wetness" in permafrost-affected
grid cells — with the meaning of wetness not being limited to the soil water content. The "DRY" setup is characterized by low
       infiltration rates, a poor soil water retention and large runoff and drainage rates, while the "WET" setup assumes higher infiltra-
       tion rates combined with a high water retention and large evapotranspiration rates. For the design of the simulations we made
       use of the optional formulations that were included in the soil hydrology scheme. In the DRY setup, a poor soil water retention
       and high drainage rates are achieved by allowing the supercooled water to move through the ground and by not impeding
the lateral drainage in the presence of ice in the subjacent soil layers (Tab. 1). Low infiltration rates were obtained by setting
       $F_{ARNO}$ to 0.8 and, with respect to evapotranspiration, a high resistance was assumed by including $\Delta\Phi_{org}$ in $W^*_{MAX,top}$ and
       $W^*_{MAX,rz}$. Finally we use a comparatively low maximum wetland-lag ($P_{lag}$=100 d) which limits the extent of inundated areas
       and the corresponding infiltration and evaporation rates. For the WET setup, we assumed high infiltration rates, allowing all
       of the reservoir outflux to be separated into infiltration and surface runoff by the ARNO scheme ($F_{ARNO}$ = 1). Furthermore,
we set a high water retention in permafrost affected soils, by assuming that supercooled water is stationary and that the lateral
       drainage flux is impeded by the ice content of the subjacent soil layers. With respect to evapotranspiration, we assume a low
       resistance by not accounting for $\Delta\Phi_{org}$ in $W^*_{MAX,top}$ and $W^*_{MAX,rz}$. We also assumed a larger maximum lag for the surface
       water bodies ($P_{lag}$=150 d) resulting in higher evaporative fluxes from inundated areas. Finally, the minimum root depth was
       increased from 10 cm to 30 cm, increasing the plant water availability mainly in the mountainous regions in eastern Siberia
(Fig. 3).

As discussed in Sec. 2.1.1 - 2.1.4, many of the above parameters and optional formulations are a means to representing the
       uncertainty resulting from the complexity of interactions and processes or from structural shortcomings of the model rather
       than the uncertainty in the specific parameter values. Thus, they are used for model tuning, even if the specific parameter
or formulation itself constitutes a measurable quantity. Furthermore, even if the parameter could at least in theory be better
       constrained, observation-based information suitable for the resolution of the model may not exist. A good example for this
       is the maximum wetland lag, which has been determined for specific lakes (Ambrosetti et al., 2003; Brooks et al., 2014)
       but upscaling the observation-based values to the model scale requires highly uncertain assumptions about the representative
       bathymetry of wetlands, ponds and lakes and their relative fraction in the overall inundated area. With the number of available
tuning options, the design of the setups is — to a certain degree — arbitrary. Here, the differences between the WET and the
       DRY simulation do not cover the complete uncertainty-range included in the parametrizations of JSBACH's soil hydrology
       scheme. For example, a much dryer setup could have been obtained by keeping JSBACH's original formulation of prohibiting
       infiltration at sub-zero temperatures or maintaining a fixed maximum root zone soil moisture — that is not reducing $W_{MAX,rz}$





by the ice-content. Furthermore, we focused on the representation of processes and, besides the minimum root depth, made
no diverging assumptions with respect to the soil properties, which are particularly uncertain in regions with large soil organic
matter concentrations. Our aim was not to compare the most extreme scenarios but setups whose differences are in the range of
"typical" inter-model differences. A measure for the latter was derived from the simulations of the Permafrost Carbon Network
Model Intercomparison Project (PCN-MIP; McGuire et al., 2018), which targets the behaviour of state-of-the-art LSMs in
permafrost-affected regions (Andresen et al., 2020). Here, our primary focus was the simulated partitioning of moisture fluxes
between runoff (including drainage) and evapotranspiration, as this ratio can be expected to be the parameter most relevant
for the climate feedback. Thus, we designed the setups in a way that the average differences in evapotranspiration in JSBACH
standalone-simulations with the WET and the DRY configuration do not (substantially) exceed the standard deviation of the
ensemble of PCN-MIP-participants (Fig. 4), that is roughly $0.3 \, \mathrm{mm \, day^{-1}}$ (Fig. 2b). A secondary goal was to maintain a similar
plant water availability for the same atmospheric conditions, so that differences in the vegetation cover in coupled simulations
can be fully attributed to the feedback-mediated differences in climate and not to setup-induced soil moisture differences. The
agreement with observations was not taken into account in the design of the setups. Nonetheless, we conducted a brief model
evaluation — focused on the northern permafrost regions — which is included in the supplementary materials (Tabs. ST1,ST2
and Figs. SF1-SF9).

### 2.3 Simulations

The general setup of the simulations uses a 450-second timestep for the atmosphere and land components, while the ocean
model is run at a 2700-second timestep. The horizontal resolution in the atmosphere and over land is T63 ($1.9° \times 1.9°$) —
which corresponds to a grid-spacing of about 200 km in tropical latitudes — and GR15 ($1.5° \times 1.5°$) in the ocean, correspond-
ing to 160 km in the tropics. The atmosphere has a vertical resolution of 47 levels reaching up to 0.1 hPa, which is a height
of about 80 km, while the ocean model uses 41 vertical layers reaching to a depth of up to 5000 m. The land is resolved by 18
subsurface layers that extend to a depth of about 160 m, 11 of which are used to represent the top 3 m of the soil column. This
is very different to the standard vertical setup which represents the soil column by 5 layers reaching to a depth of less than
10 m. Imposing a deeper bottom boundary is important for a realistic representation of the soil thermodynamic regime, with
implications for subsurface heat conduction and energy distribution (MacDougall et al., 2008; González-Rouco et al., 2009;
González-Rouco et al., 2021), as too shallow LSMs alter the distribution of temperatures in the subsurface (Alexeev et al.,
2007; Smerdon and Stieglitz, 2006). As shown by Steinert et al. (2021b) a depth > 150 m is required to resemble an infinitely
deep soil in climate-change simulations of centennial timescales. The improved vertical resolution and bottom boundary condi-
tion depth used herein produce changes in the subsurface thermal state that have been shown to interact with the phase changes
and other hydrological features. In regions with active layer dynamics, the more realistic conduction of heat from the surface
into the soil impacts the depth of the zero annual amplitude of ground temperatures, which has impacts on the simulation of
near-surface permafrost extent in the Arctic and subarctic regions (Steinert et al., 2021a).





Both simulations start in the year 1800, with the atmosphere and the ocean being initialized using pre-industrial control simulations with the standard MPI-ESM, which were performed for the CMIP6 DECK experiments. However, the land surface cannot be initialized in the same manner, as the CMIP6 DECK experiments use the standard setup. Thus, they do not include

some of the essential variables in permafrost regions — such as the soil ice content — and have a different vertical discretization in the soil. Furthermore, the states that the standard model version simulates in the permafrost regions differ substantially from those simulated with either the WET or the DRY setup. Instead, we used a temperature approximation — based on the surface temperature — to initialize the soil temperature and assumed all soils layers to be close to saturation — that is at 95% of the field capacity. Starting from this state, we ran JSBACH in standalone-mode for 200 years, with the atmospheric forcing

data derived from the MPI-ESM pre-industrial control simulations. During this period, the soil temperatures and soil water and ice content adjust to the prescribed atmospheric conditions, allowing us to use the final states to initialize the coupled simulations.

In the simulations, the water- and energy cycles of the MPI-ESM-components are fully coupled. However, this is not the case

for the biogeochemical cycle. Especially the magnitude of the permafrost carbon feedback is extremely difficult to estimate. Accounting for it in our investigations would not only have required extensive adaptations in JSBACH, but also to perform a number of ensemble-simulations to account for the uncertainty that is included in the formulations of the permafrost carbon cycle (de Vrese et al., 2021; de Vrese and Brovkin, 2021). The latter increases the computational demand of the experiment by an order of magnitude, while not necessarily providing any additional insights into the physical land-atmosphere feedbacks

in permafrost regions, which are the main focus of this investigation. Instead, we ran the model with prescribed atmospheric greenhouse gas concentrations, which corresponds to the assumption that present-day and future atmospheric $CO_2$ levels are largely determined by human activity and that all variations in the natural carbon fluxes are offset by corresponding adjustments in the anthropogenic emissions. The simulations start with an atmospheric $CO_2$ concentration of about 280 ppmv, which increases to about 400 ppmv between the years 1800 and 2014. After 2014, the simulations follow a high GHG emission

trajectory based on the Shared Socioeconomic Pathway 5 and the Representative Concentration Pathway 8.5 (SSP5-8.5; van Vuuren et al., 2011). SSP5-8.5 targets a radiative forcing of 8.5 W m$^{-2}$ in the year 2100 and assumes the atmospheric $CO_2$ concentrations to increase to about 1100 ppmv. We chose to investigate SSP5-8.5, even though it may not be be based on the most plausible assumptions (van Vuuren et al., 2011; Riahi et al., 2017; Hausfather and Peters, 2020), but investigating extreme scenarios often helps in highlighting impacts and understanding causal relationships.


Furthermore, we conducted an additional set of simulations in which the atmospheric greenhouse gas concentrations were prescribed in a way as to stabilize the global mean surface temperature at 1.5°C above the pre-industrial CMIP6 ensemble mean temperature. For the DRY setup this required lowering atmospheric $CO_2$ from 450 ppmv in the year 2030 to 400 ppmv in the year 2080. For the WET setup $CO_2$ levels were reduced from 530 ppmv to 465 ppmv between the years 2045 and 2095.

It should be noted that the differences between these two simulations are not exclusively a result of the differences in model parametrizations but also of the differences in the strength of the $CO_2$ fertilization effect. Without dedicated simulations,



excluding the effect of rising atmospheric $CO_2$ on vegetation, it is not possible to determine how important the additional 68 ppmv in the WET simulation are for the simulated vegetation covers and the resulting feedbacks on the climate system. However, the WET simulation shows a slightly smaller overall vegetation cover at higher $CO_2$ and the same global mean
temperature, indicating that the additional $CO_2$ fertilization is less important than the differences in the spatial distribution of temperatures and precipitation.

## 2.4 ICON-ESM

The present investigation relies exclusively on simulations with MPI-ESM. However, to confirm that our findings do not merely describe a specific feature of this particular model, but provide more general insights, we conducted an additional set of
simulations with the new ICON-Earth System Model (ICON-ESM; Jungclaus et al., 2022). The ICON-ESM is the first ESM using the unstructured, icosahedral grid concept of the ICON framework. It consists of the atmospheric component ICON-A, with a non-hydrostatic dynamical core, the ocean model ICON-O, which builds on the same ICON infrastructure but applies the Boussinesq and hydrostatic approximation, and the ICON-L land module. The latter provides a new framework with a flexible scheme of land surface tiling and an object-oriented organization of physical and biogeochemical processes, which
allows the coupling of different land surface schemes to the atmospheric and ocean components. For the present investigation we used the land surface model JSBACH4, the sucessor of JSBACH3.2, the land surface component of the latest MPI-ESM version (Mauritsen et al., 2019). The model was adapted as described in section 2.1, allowing to simulate dry or wet conditions in the northern permafrost regions. However, as the WEED scheme could not be implemented into JSBACH4 yet, the model does not represent the ponding of water at the land surface and the evapotranspiration from surface water bodies. Furthermore,
the natural vegetation dynamics have not yet been enabled in JSBACH4, and neither the changes in plant water availability resulting from the modifications of the permafrost hydrology nor the increase in the atmospheric $CO_2$ concentrations and in the near-surface temperatures have an effect on the vegetation distribution in the high latitudes. Because of this limited functionality of JSBACH4, we do not include the simulations in the main analysis of this study and merely show the most important results in the supplementary materials (Figs. SF10 - SF12). Overall, the results obtained with ICON-ESM agree
well with those of the MPI-ESM simulations, despite ICON-ESM being run at a comparatively low horizontal resolution, i.e. R2B3, in which the globe is resolved by 5120 triangular cells with an average distance between cell centers of 277 km, with the standard 5-layer vertical resolution in land cells, and despite the model simulating a dryer climate in the Acrtic and susbarctic zone.

## 2.5 Uncertainty in Arctic and subarctic climate projections

With evapotranspiration rates being partly predetermined — as one of the target variables in the design of our setups (see "Setups" above) — we focus on the uncertainty in simulated surface temperatures and precipitation and how these relate to the evapotranspiration rates. In the northern permafrost regions, the simulations of the CMIP6 ensemble (Tab. 2) show a high inter-model correlation between precipitation and evapotranspiration — that is a Spearman's $\rho$ of 0.89 with a p-value of 1.88 $\times 10^{-15}$ for the average over the permafrost-affected grid cells and over the period 2000 - 2100 (not shown). But, while there





appears to be little uncertainty that the two processes are closely connected, it is not clear whether high (low) precipitation rates in a model are caused by high (low) evapotranspiration rates or vice versa, or even whether there is little causal relation between the two but a similar dependency on a third factor. Here, our investigation aims to establish or eliminate soil-setup induced differences in evapotranspiration rates as a potential cause of the variations in high-latitude precipitation between the CMIP6 participants. In contrast, the CMIP6 ensemble exhibits a low correlation between the average surface temperatures

and evapotranspiration rates — that is a $\rho$ of 0.28 with a p-value of 0.07. The poor correlation indicates that the surface temperatures are not exclusively determined by the evapotranspiration rates but by a number of factors — one of which may be evapotranspiration. Here, our simulations offer the chance to isolate the potential contribution of permafrost hydrology related evapotranspiration differences to the uncertainty in surface temperature in climate change projections.

As a measure for the uncertainty of these projections, we use the inter-model spread of the CMIP6 ensemble, which was derived from 44 simulations provided by 27 modelling groups (Tab. 2). Here, it should be noted that we use the term "inter-model" in a loose sense, as the ensemble does not consist of simulations with 44 different ESMs, but also includes multiple simulations with the same model being run with a different configuration — e.g. a high vs. a low resolution or with and without interactive vegetation. However, in the case that several simulations were available for the same model-configuration

but merely differ with respect to the initial conditions or minor parameter settings, we consider only one of the simulations to not underestimate the inter-model spread. To exclude outliers, we do not define the spread as the difference between ensemble minimum and maximum. Instead, we use two frequently applied measures, namely the interquartile range (IQR) — that is the difference between the $25^{th}$ and the $75^{th}$ percentile — as well as two ensemble standard deviations ($2\sigma$) — with the simulations within one standard deviation of the mean covering about two thirds of the ensemble, if the latter is normal distributed.

## 3 Results

### 3.1 Land-Atmosphere interactions

One of the questions that motivated this investigation is whether the Arctic and subarctic zone will become wetter or dryer in response to future warming, or rather if present-day ESMs can be used to predict this with some degree of confidence, given the uncertainties in the hydrology-parametrizations of the land surface components. Both the WET and the DRY simulation

predominantly show an increase in evapotranspiration and precipitation, even though the signal is not uniform across the northern permafrost areas (Fig. 5a,b,d,e). Especially in North America there are extensive regions in which the evapotranspiration rates decline during the 21$^{st}$ century. These regions enclose most of the areas that also exhibit a negative trend in precipitation, indicating that the latter is the result of a reduced moisture recycling. Nonetheless, the general trend is an increase in the intensity of the hydrological cycle with the signal being more pronounced in WET than in DRY. In contrast, the two sim-

ulations show opposing trends in the total soil water content — that is liquid water and ice (Fig. 5c,f). In WET, the soils in most grid cells lose water, with the trend being partly driven by a strong increase in the drainage rates (not shown). In this setup, lateral drainage is strongly inhibited in the presence of ice and the marked increase in these fluxes is the result of the





warming-induced decrease in the soil ice concentration. In the DRY setup, the inhibition of the lateral subsurface flow in the presence of ice is less severe, hence, the increase in drainage due to permafrost degradation is less pronounced. As a result, the soils in the DRY simulation almost exclusively exhibit an increase in the water content, with the increase in precipitation (-evapotranspiration) being the dominant signal. It should be noted that JSBACH, as most land surface models, does not include a representation for excess ice. Excess ice is the water that the soil can only hold when frozen, but which exceeds the pore volume of the unfrozen ground. Thus, the thawing process effectively reduces the amount of water that the soils can hold and it is plausible that the increase in the water content found in the DRY setup — but also found in other models (Andresen et al., 2020) — is only possible because the model neglects the feature of a higher soil water holding capacity in the frozen state.

Thus, when considering the total soil water content to be a key indicator, the uncertainty in the representation of the permafrost hydrology makes it impossible to provide an unambiguous answer to the question whether the Arctic and subarctic region will become wetter or drier in the future. Furthermore, the agreement on the direction of the trends in evapotranspiration and precipitation does not mean that the different soil hydrology setups entail similar conditions at and above the surface. On the contrary, the land-atmosphere interactions diverge substantially between WET and DRY, which has a distinct impact on the near-surface climate. In WET, the evapotranspiration rates are between 0.2 and 0.3 mm day$^{-1}$ larger than in DRY, which translates into a difference in latent heat flux of about 5 W m$^{-2}$ at the beginning and about 9 W m$^{-2}$ at the end of the 21$^{st}$ century (Fig. 6a). The additional evaporative cooling in WET constitutes 6 % to 10 % of the shortwave radiation that is absorbed by the surface and profoundly changes the partitioning of the surface energy budget. With the Bowen ratio decreasing to 0.3, the permafrost-affected areas are increasingly energy limited in WET, while in DRY, a Bowen ratio of almost 1 indicates that at least parts of the region experience some degree of water limitation (Fig. 6b).

With less sensible- and more latent heat being transferred into the atmosphere, the boundary layer is initially cooler and moister in WET than in DRY (not shown). This leads to a higher relative humidity and, consequently, to precipitation rates that are roughly 0.2 mm day$^{-1}$ larger in WET (Fig. 6c). The higher precipitation rates in turn increase the soil water availability, establishing a positive feedback in which the more intense moisture recycling is the main factor sustaining the higher evapotranspiration rates. More specifically, about 80 % of the additional evapotranspiration in WET are compensated for by higher precipitation rates, while merely 20 % are balanced by differences in runoff (Fig. 6d). Furthermore, the differences in relative humidity result in differences in the cloud cover which constitute another important feedback on the surface energy balance (Fig. 6e). In WET, the increased cloudiness raises the planetary albedo (relative to DRY), reducing the surface incoming solar radiation by between 10 W m$^{-2}$ at the beginning and 13 W m$^{-2}$ at the end of the 21$^{st}$ century. When additionally taking into consideration the differences in the surface reflectivity — resulting from differences in the simulated snow and vegetation covers — the differences in absorbed shortwave radiation amount to roughly 12 W m$^{-2}$ (Fig. 6f). The reduction of the available energy cools the surface further, which leads to less sensible heat being transferred into the boundary layer, contributing to the higher relative humidity in the atmosphere. Here, it should be noted that the differences in evaporative cooling and the planetary albedo affect the available energy very differently, because the former mainly redistribute, while the latter provide a

 

net change to the energy content of the system. However, even when focusing exclusively on the surface latent heat flux and the incoming shortwave radiation, the cloud effect (10 W m$^{-2}$ - 13 W m$^{-2}$) has a larger impact on the surface energy balance
than the evaporative cooling that initially caused it (5 W m$^{-2}$ - 9 W m$^{-2}$).

The energy balance determines the temperatures at and below the surface, which in turn are highly relevant for the question whether the high latitudes will become wetter or dryer in the future. It can be argued that a more suitable measure for the wetness of the permafrost areas is the liquid- rather than the total water content of the soils, with the former controlling
the majority of the physical and biophysical land processes. Here, the trend in liquid soil moisture does not only depend on the development of the total water content, but also on the temperature evolution, as these determine the ratio of liquid and frozen water in the soil (Fig. 6g). In both simulations, the 21$^{st}$-century warming results in a substantial decline in the extent and thickness of the near-surface permafrost (Fig. 6h), leading to a marked decrease in the soil ice content and a corresponding increase in liquid water (Fig. 6i). Thus, when basing the question of future wetness on the liquid soil water
content, our simulations provide a clear answer: The trends in evapotranspiration, precipitation and soil moisture all suggest that, on average, the Arctic and subarctic region will become wetter in the future. This general trend appears to be a robust feature, the direction of which is independent of the representation of the permafrost hydrology in the model. However, the magnitude of the (liquid) soil moisture trend is very sensitive to the parametrization of the model, mainly to the resulting differences in the simulated surface temperatures. Here, the WET simulation is consistently colder, which results in a higher
near-surface permafrost volume at the beginning of the 21$^{st}$ century and a lower liquid soil moisture content. With the higher initial permafrost volume, the 21$^{st}$-century warming has a more pronounced impact on the thaw rates, and the resulting trend in the liquid soil moisture is much larger in WET than in DRY, despite WET exhibiting a negative trend in the total soil moisture (Fig. 5c,f; Fig.6i).

### 3.2   Differences in climate compared to the CMIP6 spread

Another question motivating our study was to which extent differences in the parametrizations of the soil hydrology could help explain the large inter-model spread that the present generation of ESMs exhibits in the Arctic and the subarctic zone. In the northern permafrost regions, the absolute value of the differences in evapotranspiration between DRY and WET ($\Delta^{evap}_{|DRY-WET|}$) captures the range of "typical" inter-model differences comparatively well and at the beginning of the 21$^{st}$ century they match the respective interquartile-range of the CMIP6 ensemble (IQR$^{evap}$) almost perfectly (Fig. 7a). Subsequently, $\Delta^{evap}_{|DRY-WET|}$
increases considerably, exceeding IQR$^{evap}$ by 2030, but remains well within the range of two (CMIP6) ensemble standard deviations (2$\sigma^{evap}$). It should be noted that this good agreement was to be expected as the setups were designed to produce differences that don't exceed the range of "typical" evapotranspiration differences between commonly used LSMs (although these "typical" differences were not determined based on fully coupled CMIP6 simulations, but on the ensemble of standalone simulations that were performed for the PCN-MIP). Thus, the more interesting question is how sensitive the simulated climate
is to these differences in evapotranspiration, more specifically whether the latter lead to differences in other key variables that





are similarly consistent with the respective CMIP6 spreads.

In the case of precipitation, the differences in the soil-hydrology parametrizations offer a large explanatory potential and $\Delta^{pr}_{|DRY-WET|}$ matches $\mathrm{IQR}^{pr}$ for most of the 21$^{\text{st}}$ century (Fig. 7b). On average $\Delta^{pr}_{|DRY-WET|}$ amounts to about $0.16\,\mathrm{mm\,day}^{-1}$,

$\mathrm{IQR}^{pr}$ to about $0.19\,\mathrm{mm\,day}^{-1}$ and $2\sigma^{pr}$ to $0.32\,\mathrm{mm\,day}^{-1}$. Thus, even if considering $2\sigma^{pr}$ to be the more appropriate measure, about half of of the inter-model spread of the CMIP6 ensemble may be explainable by diverging evapotranspiration rates resulting from differences in the parametrizations of the permafrost hydrology. Both $\mathrm{IQR}^{pr}$ and $\Delta^{pr}_{|DRY-WET|}$ remain comparatively constant during the 21$^{\text{st}}$ century , while $2\sigma^{pr}$ clearly exhibits a positive trend. Thus, the question how well $\Delta^{pr}_{|DRY-WET|}$ captures the temporal development of the "typical" inter-model differences depends partly on the measure for the spread. In regards

to the surface temperatures, $\Delta^{ts}_{|DRY-WET|}$ matches the magnitude of the CMIP6 ensemble spread similarly well, although the issue of the 21$^{\text{st}}$-century trend is even more prominent (Fig. 7c). In the northern permafrost regions, the average $\Delta^{ts}_{|DRY-WET|}$ is equal to $\mathrm{IQR}^{ts}$ (2.6 °C) and represents about two thirds of $2\sigma^{ts}$ (3.7 °C). Here, $\Delta^{ts}_{|DRY-WET|}$ shows a good agreement with the inter-model spread during the first half of the century, ranging between $\mathrm{IQR}^{ts}$ and $2\sigma^{ts}$. However, during the second half of the century, both $\mathrm{IQR}^{ts}$ and $2\sigma^{ts}$ increase by more than 1°C, while $\Delta^{ts}_{|DRY-WET|}$ shows no significant increase. Thus,

$\Delta^{ts}_{|DRY-WET|}$ matches the magnitude of the CMIP6 spread well, but appears to lack an important dynamical component (see below). Non-glaciated, permafrost-affected areas only cover about a third of the planet's surface north of 50°N. Yet the average $\Delta^{ts;+50N}_{|DRY-WET|}$ in these latitudes — including glaciers and the ocean — amounts to about 2.0 °C (Fig. 7d). This is notably less than $\Delta^{ts}_{|DRY-WET|}$, but still more than twice as much as would have been the case if the temperature effects were limited to the land areas affected by permafrost. This clearly shows that the differences in the permafrost hydrology have an impact on

the entire Arctic and subarctic region and may still explain a large part of the respective inter-model spread — in comparison $\mathrm{IQR}^{ts;+50N}$ averages about 2.7 °C and $2\sigma^{ts;+50N}$ 3.8 °C. However, both $\mathrm{IQR}^{ts;+50N}$ and $2\sigma^{ts;+50N}$ increase by more than 2 °C in the course of the 100 year period while, again, $\Delta^{ts;+50N}_{|DRY-WET|}$ shows no noteworthy trend.

It may appear counterintuitive that the temperature differences between WET and DRY remain fairly constant during the

21$^{\text{st}}$ century despite the difference in evapotranspiration and evaporative cooling increasing over time. The reason for this is that evapotranspiration initially lowers the temperatures at the surface, but it eventually increases the air column temperature by the heat release during condensation. Thus, the evaporative cooling redistributes energy between latent and sensible heat and between surface and atmosphere, but does not change the energy content of the coupled land-atmosphere system. The combination of lower surface temperatures and higher atmospheric temperatures, in turn, increases the downward fluxes of

sensible heat and long-wave radiation or decreases the upward fluxes, which largely balances the effect of the evaporative cooling at the surface. Over longer periods, local surface temperatures only change significantly if the evaporated or transpired water is advected out of the region and the net effect can be approximated by the latent heat included in the precipitation - evapotranspiration difference (P-E). Here, WET and DRY exhibit similar trends — as indicated by the trends in surface runoff and drainage (Fig. 6d; note that over longer periods P-E ~ runoff & drainage) — and the difference in P-E remains constant over

time. Furthermore, the divergence of the evapotranspiration rates results in increasingly large differences in the cloud cover,





affecting the planetary albedo. In contrast to the evaporative cooling, the albedo differences change the amount of energy reflected back to space, hence the total energy content of the system. However, the albedo effects resulting from the divergence in the cloud cover are compensated for by decreasing differences in the surface reflectivity, mainly resulting from a convergence of the simulated vegetation covers. As a result, the differences in absorbed shortwave radiation do not change over time (Fig.

6f) and, with no significant trends in the differences neither in P-E nor in the planetary albedo, the temperature differences between WET and DRY remain almost constant during the 21$^{st}$ century.

This raises the question whether the uncertainty in the permafrost hydrology cannot help to explain the increase in the inter-model spread, especially during the second half of the century, or whether the issue of constant temperature differences is

specific to our setups. Per design, the simulations with the WET and the DRY setup become more similar over time as many of the distinctions depend on the soil ice content, which decreases throughout the 21$^{st}$ century. To test how far the lack of trends is related to this design-feature, we compared the WET simulation to a simulation using a third, highly synthetic setup — the W2D setup — which is based on dynamical parametrizations in the northern permafrost regions. This setup assumes that the characteristics of the soil hydrology are determined by the presence of near-surface permafrost and change when the

latter is degraded. All grid cells start with the parametrizations of the WET setup and the configuration is maintained as long as the model simulates permafrost in the upper 3 m of the soil. However, the parametrizations switch from WET to DRY (with the exception of the soil depths and the maximum wetland retention $P_{lag}$) whenever the annual maximum thaw depth in the grid-cell extends beyond a depth of 3 m. For the high-emission scenario considered in this study, the majority of the grid cells in the northern permafrost regions transitions from WET to DRY during the 21$^{st}$ century, with the W2D simulation becom-

ing increasingly different from the WET simulation. Thus, $\Delta^{pr}_{|W2D-WET|}$ exhibits a trend that is substantially larger than the trends in $\Delta^{pr}_{|DRY-WET|}$ and IQR$^{pr}$, closely matching the trend in $2\sigma^{pr}$ (Fig. 8b). Furthermore, the trend of $\Delta^{ts}_{|W2D-WET|}$ in permafrost regions is very similar to the trend in IQR$^{ts}$ (Fig. 8c) and also north of 50°N the trend in $\Delta^{ts;+50N}_{|W2D-WET|}$ increases fivefold relative to $\Delta^{ts;+50N}_{|DRY-WET|}$, amounting to half of the trend in $2\sigma^{ts;+50N}$ (Fig. 8d). This indicates that the lack of trends in $\Delta^{ts}_{|DRY-WET|}$ and $\Delta^{ts;+50N}_{|DRY-WET|}$ is at least partly design-related. However, the trends stem from a much larger causative

trend in $\Delta^{evap}_{|W2D-WET|}$. The latter is almost twice as large as the trend in $2\sigma^{evap}$ and almost 20 times larger than the trend in IQR$^{evap}$ (Fig. 8a), strongly suggesting that the trends in the temperature-spread of the CMPI6 ensemble are not caused exclusively by the divergence of evapotranspiration rates.

## 4 Conclusions

Even though our simulations show that the differences in the parametrization of the permafrost hydrology may not fully explain the spread of the CMIP6 ensemble, $\Delta^{ts}_{|DRY-WET|}$ and $\Delta^{ts;+50N}_{|DRY-WET|}$ still represent sources of large uncertainty in the temperatures of the Arctic and subarctic region. These temperatures constitute the main drivers of a number of important processes, some of which have implications for the global climate. For example, the magnitude of the permafrost carbon feedback



is largely determined by the speed with which the vast pools of soil organic matter in the Arctic and subarctic zone will be-
come exposed to conditions that are required for microbial decomposition, hence on the rate of permafrost thaw. As shown
above, the simulated degradation of the terrestrial permafrost is very different in WET and DRY and the point in time when the
near-surface permafrost (almost) disappears from the northern high latitudes differs by about 50 years (Fig. 9). The terrestrial
net carbon flux in the Arctic and subarctic region also depends on the trend in Arctic greening, as the expanding vegetation
takes up increasingly large amounts of atmospheric $CO_2$ (Qian et al., 2010; Keenan and Riley, 2018; Pearson et al., 2013;
McGuire et al., 2018; Zhang et al., 2018). Here, the surface temperatures determine the length of the growing season, with
$\Delta^{ts}_{|DRY-WET|}$ leading to substantial differences between the simulated vegetation dynamics in WET and DRY. For example,
the moment after which the tree cover exceeds a third of the land surface in the permafrost-affected regions differs by more
than 60 years between the two simulations. Both processes, permafrost degradation and Arctic greening, are highly relevant
for the atmospheric greenhouse gas concentration but are not the only ways for the permafrost hydrology to affect the global
climate.

As discussed above, a divergence of the cloud cover is the main driver of the temperature differences between WET and
DRY. The resulting impacts on the planetary albedo are not compensated by opposing feedbacks neither in the permafrost-
affected areas nor in adjacent regions, resulting in persistent temperature differences throughout the Arctic and subarctic zone.
These temperature differences, in turn, modulate the latitudinal temperature gradient, causing a change in the meridional heat
transport which leads to a southward propagation of the temperature signal. In fact, the differences in the permafrost hydrol-
ogy impact the total energy content of the Earth system to such an extent that surface temperatures across the entire Northern
Hemisphere are significantly affected. Thus, the global mean temperature in WET and DRY differs by about 0.5 °C and 0.6 °C
at the beginning and the end of the 21ˢᵗ century, respectively (Fig. 9), despite the non-glaciated, permafrost-affected areas in
the Arctic and subarctic region covering merely 5 % of Earth's surface. Here, 0.5 °C - 0.6 °C constitute substantial differences,
not only in comparison to the CMIP6 ensemble spread, but also relative to the temperature increase during 21ˢᵗ century. In the
case of the high-emission scenario SSP5-8.5, the point in time when the simulations reach the same global mean temperature
differs by about 15 years (Fig. 9). And because the albedo-differences affect primarily the Arctic and subarctic region, this
global mean temperature is reached with a substantially different latitudinal distribution (Fig. 10a).


The sustainability of a given climate trajectory depends on the associated risks for natural and human systems (IPCC, 2018),
some of which stem from regional tipping elements (Lenton et al., 2019), such as the West African monsoon, reaching a crit-
ical threshold. How close these elements are to a tipping point at a given global mean temperature depends on the latitudinal
temperature gradient which determines the local temperature change. For a climate stabilization at a desirable level — e.g.,
1.5 °C above the pre-industrial mean — the state of many tipping elements in the northern cryosphere is distinctly different
between DRY and WET. The near surface-permafrost volume in the northern high latitudes is 15 % - 30 % lower in DRY than
in WET and the ablation rates of the Greenland ice sheet differ by up to 1 mm day$^{-1}$. In addition, the annual mean Arctic sea-
ice concentration is reduced by up to 15 % (Fig. 10b,c) in the DRY simulations. The difference in the simulated ice coverage



is particularly prominent during the summer months with the DRY simulation featuring an almost ice-free Arctic ocean, while
around $2 \times 10^6$ km$^2$ remain ice-covered in the WET simulation (not shown). In the SSP5-8.5 simulations, the differences during summer are even more pronounced, with the DRY simulations reaching an ice-free state several decades before the WET simulations. These findings, again, shed an interesting light on the regional differences of the temperature response to the two scenarios. For a given model, and for the observational record, a clear, linear relationship between global mean temperature and Arctic sea-ice coverage has long been identified across all months. However, our investigation now shows for the first time that for a given model the same global mean temperature, in our case a warming of +1.5 °C, can result in differences in the simulated sea-ice coverage owing to differences in the regional amplification of the global temperature signal. Our simulations also show that these differences are not necessarily equally pronounced across all months, as the sea-ice concentration in March in the SSP5-8.5 simulations are barely distinguishable between the WET and the DRY simulations, while they are clearly different in September.

The sea-ice cover has a strong effect on the benthic temperatures, which, in turn, determine the state of the roughly $3.5 \times 10^6$ km$^2$ of permafrost soils that have been submerged since the Last Glacial Maximum and form a large part of the Arctic Shelf (Sayedi et al., 2020; Steinbach et al., 2021). The permafrost-affected sediments hold about 500 Gt of organic carbon and methane gas, with continuous thawing from the surface increasing the vulnerability of the carbon pools (Schuur et al., 2015). Here, the subsea permafrost extent near the sea bottom is strongly affected by the temperature differences between DRY and WET, in particular in the Laptev Sea and the East Siberian Sea where the frozen fraction in the top 10 m of the subsurface differs by up to 50 % (Fig. 10d).

In the boreal zone, the differences in the simulated temperatures and vegetation covers have a strong impact on the frequency and extent of wild fires. In DRY, the burned grid-cell fraction is up to 2 % year$^{-1}$ larger, reaching up to 10 times the area burned in WET (Fig. 10e). Furthermore, the strength of the Atlantic Meridional Overturning Circulation (AMOC) is partly determined by the rate with which the surface currents cool in the North Atlantic. The lower latitudinal temperature gradient in DRY reduces this part of the thermohaline circulation, weakening the AMOC by up to 1.5 Sv ($10^6$ m$^3$ s$^{-1}$) in both the North and South Atlantic Ocean (Fig. 10i). However, not all tipping elements are negatively affected by the smaller temperature gradient in DRY. Most prominently, the position and latitudinal oscillation of the Intertropical Convergence Zone is shifted, which results in a higher intensity of the West African monsoon in DRY. Precipitation rates during the period June - September increase by up to 1 mm day$^{-1}$ relative to WET, which constitutes up to 70 % percent of the precipitation in the Sahel zone (Fig. 10f). This difference in the monsoon precipitation increases the plant available water and the simulated vegetation cover is up to 15 % larger in DRY than in WET (Fig. 10h). In the Amazon basin, the precipitation rates during multi-annual periods of low precipitation (here 3 years) are up to 1 mm day$^{-1}$ larger in DRY (Fig. 10g). This corresponds to a relative difference of up to 40 % of the drought precipitation and may have implications for a potential dieback of the Amazon rainforest.



The above examples are by no-means a complete list of the remote effects resulting from the differences in the permafrost hydrology. However, they clearly show how important the respective parametrizations are for the global climate, even though the northern permafrost regions make up only a small part of the land surface. It is very difficult to judge whether one of the setups actually simulates a present-day climate that is closer to observations, which could be taken as an indicator for a better representation of the processes and may even suggest which future climate trajectory is more likely for a high-emission scenario. Here, an evaluation yields ambiguous results with none of the setups showing a better agreement with the observations for all the variables considered in the comparison (see supplementary materials Fig. SF1-SF9). For example, the WET setup simulates surface temperatures that are much closer to observations, while the DRY setup exhibits a lower bias with respect to the permafrost temperatures and precipitation rates. This ambiguity of the results does not necessarily mean that both setups are similarly ill-suited to represent the northern permafrost regions, as such a comparison can only evaluate the performance of the ESM as a whole but may be less revealing for individual components of the model. Thus, even if one of the setups provides a more realistic representation of the processes, it may increase the bias in a given variable simply by removing a compensating error. However, another possible explanation is that neither the WET nor the DRY setup is capable of representing the broad spectrum of soil conditions and processes that determine the land-atmosphere interactions across the entire region. Similar to W2D, a suitable setup may require different parametrizations in specific grid cells, which may also vary depending on the state of the soil. This, however, is very difficult to achieve in the often highly heterogeneous northern permafrost regions where "wet" and "dry" conditions may coexist in close proximity and different parts of a grid cell are better represented by either of the setups.

Here, one potential strategy is to increase the horizontal resolution of the model to a point at which the spatial heterogeneity is resolved. But, while it is feasible to run the LSM in a standalone-mode over a limited domain, using a resolution of few meters, it will likely remain impossible for quite some time to run a fully coupled ESM over longer periods with a resolution approaching the kilometer-scale. Another way to increase the "resolution" of the model is to introduce additional layers of tiling, with the added tiles representing different factors that determine the hydrological conditions and the land-atmosphere interactions. Such a tiling would need to account for the subgrid-scale variations in the soil properties but also for the numerous processes that redistribute water horizontally within the grid cell. With respect to the former, suitable data already exists for a large number of soil properties, but for other important parameters - such as the distribution of ice wedges — high-resolution data is not available on the pan-Arctic scale. For the treatment of the lateral movement of water the current generation of LSMs requires a number of new sub-modules that account for the moisture variability on the sub-meter scale, e.g. in the polygonal tundra (Cresto Aleina et al., 2013), but also for the lateral fluxes from high to low-lying areas along gradient slopes that act on the scales of tens to thousands of meters (Nitzbon et al., 2021; Smith et al., 2022). Given that this will require at least one additional layer of tiles and that the hydrological conditions may vary over short periods of time, which results in a comparatively fast changes in the tile fractions, this approach requires a flexibility in the model structure, which few of the present-day LSMs possess.



*Code and data availability.* The primary data is available via the German Climate Computing Center long-term archive for documentation data (https://will-be-optained-at-a-later-stage-of-the-review-process.de). The model, scripts used in the analysis and other supplementary information that may be useful in reproducing the authors' work are archived by the Max Planck Institute for Meteorology and can be
obtained by contacting publications@mpimet.mpg.de.

*Author contributions.* PdV,JFGR,DN,TS,NJS,SW,VB designed experiment. PdV, TS, SW performed simulations. PdV, GG, DN, VB conducted analysis and validation of simulations. All authors contributed to and reviewed the manuscript.

*Competing interests.* The authors declare that they have no competing financial interest.

*Acknowledgements.* This work was funded by the German Ministry of Education and Research as part of the KoPf-Synthese project (BMBF
Grant No. 03F0834C), by the German Research Foundation as part of the CLICCS Clusters of Excellence (DFG EXC 2037) and by the European Research Council (ERC) under the European Union's 691 Horizon 2020 research and innovation program (grant agreement No 951288, Q-Arctic).



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



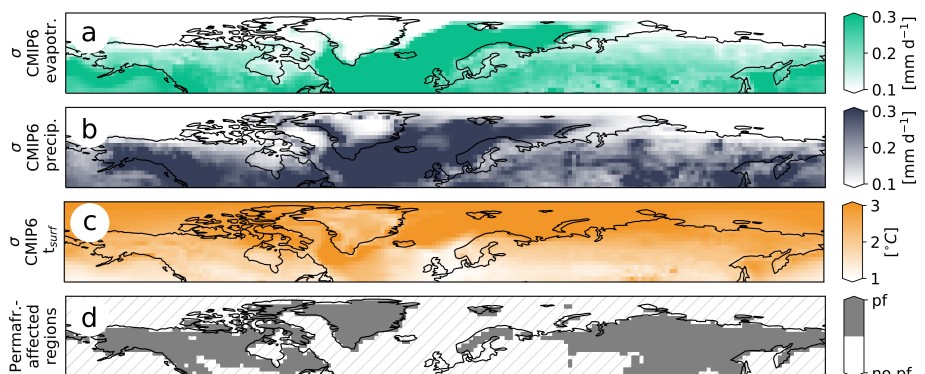

**Figure 1. CMIP6 ensemble spread and permafrost regions:**
Standard deviation ($\sigma$) of the CMIP6 model ensemble averaged for the period 1980 - 2000: a) Evapotranspiration, b) precipitation and c) surface temperatures. Plots are based on 44 simulations provided by 27 modelling groups (see Tab. 2). d) Northern mid- and high-latitude permafrost regions.



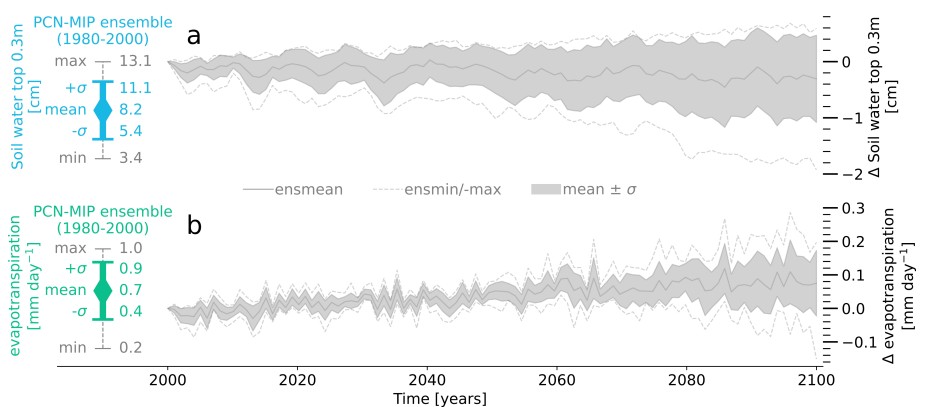

**Figure 2. PCN-MIP:**

a) Simulated soil water content in the top 30 cm of the soil averaged across the northern permafrost regions. Shown are the minimum, the mean (+/- one standard deviation; $\sigma$) and the maximum of the ensemble of Permafrost Carbon Network Model Intercomparison (PCN-MIP; McGuire et al., 2018) participants. The left side shows the 1980-2000 mean and the right side the changes during the 21[st] century (relative to the 1980-2000 mean). b) Same as a) but for evapotranspiration. The figure is based on Andresen et al. (2020), but the analysis was slightly modified in that we aggregated the output of all models over the permafrost regions shown in Fig. 1d), instead of the initial permafrost domain as simulated by the individual models. Furthermore, we included a simulation with the JSBACH model in the intercomparison.





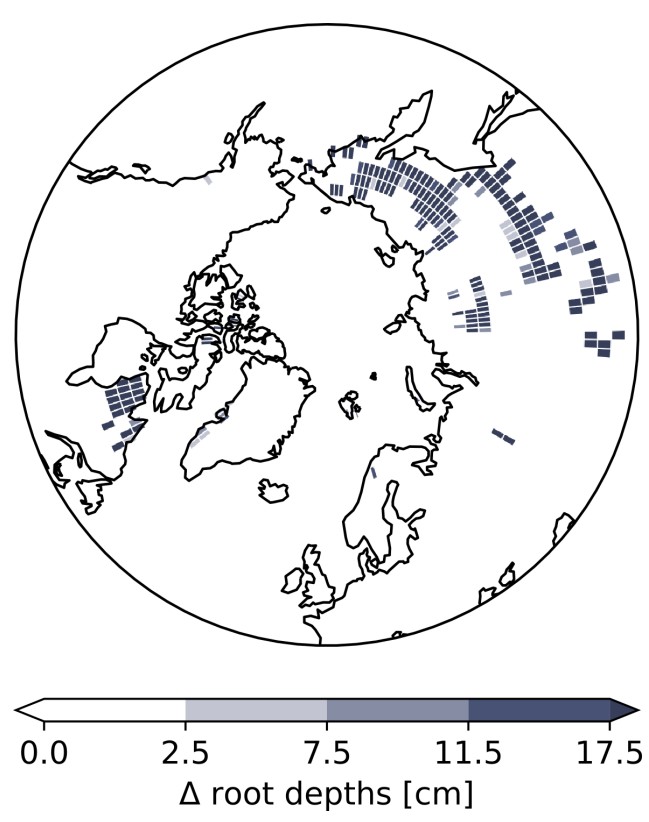

**Figure 3. Root depths**
Difference in the maximum root depths between the WET and the DRY setup.

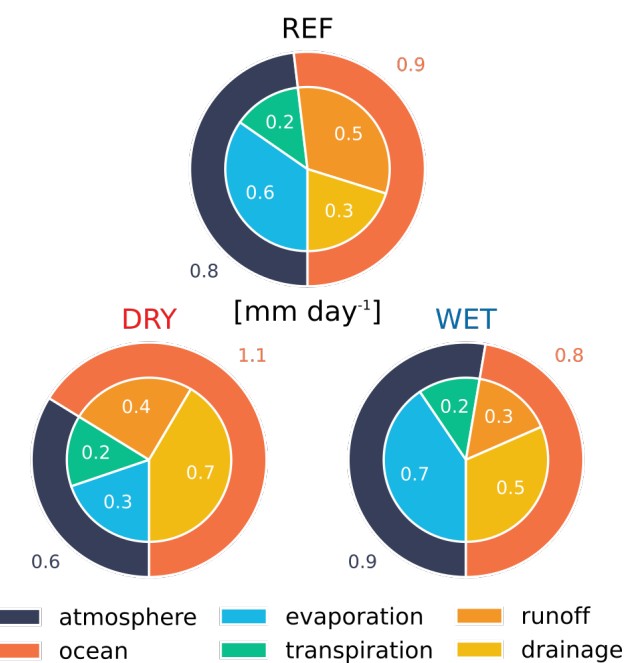

**Figure 4. JSBACH setups:**
Partitioning of the outgoing moisture fluxes amongst fluxes to the atmosphere and to the ocean (via the river discharge) and further subdivision into evaporation, transpiration, runoff and drainage. Figures show the average over the northern permafrost regions taken from standalone simulations with the atmospheric forcing corresponding to pre-industrial control conditions. The sum of all outgoing fluxes is equal to the precipitation rates, which is the same in all simulations (1.7 mm day$^{-1}$). Shown is the partitioning for the reference model (top), the DRY (left) and the WET setup (right).


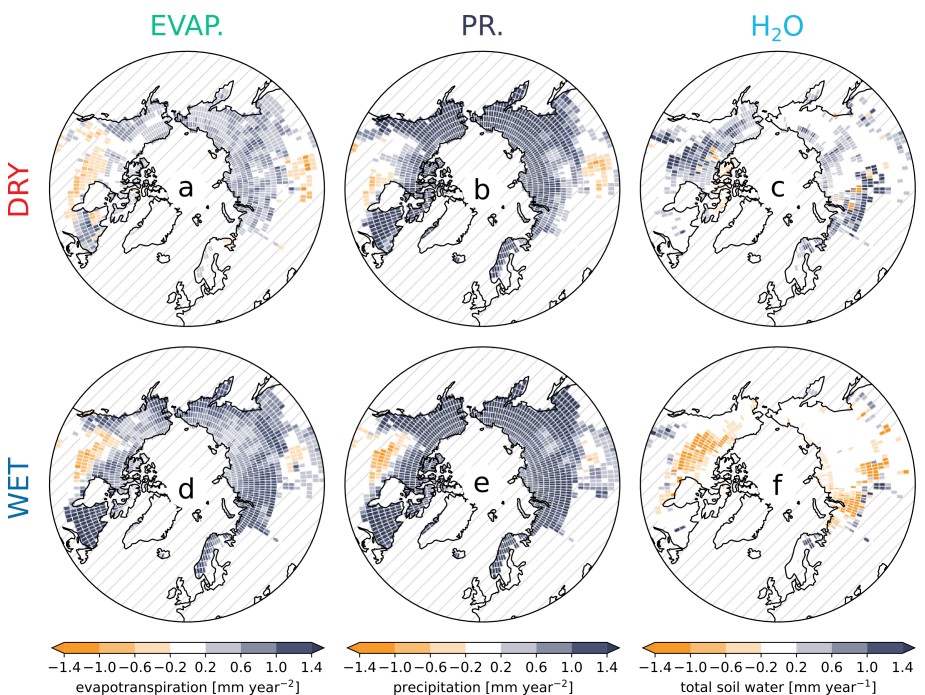

**Figure 5. Arctic futures**

a) 21$^{st}$ century evapotranspiration trend in the DRY simulation. b) Same as a) but for precipitation. c) Same as a) but for the total soil water (liquid soil moisture and ice) content. d,e,f) Same as a,b,c) but for the WET simulation. Non-permafrost and glacier grid cells are hatched.



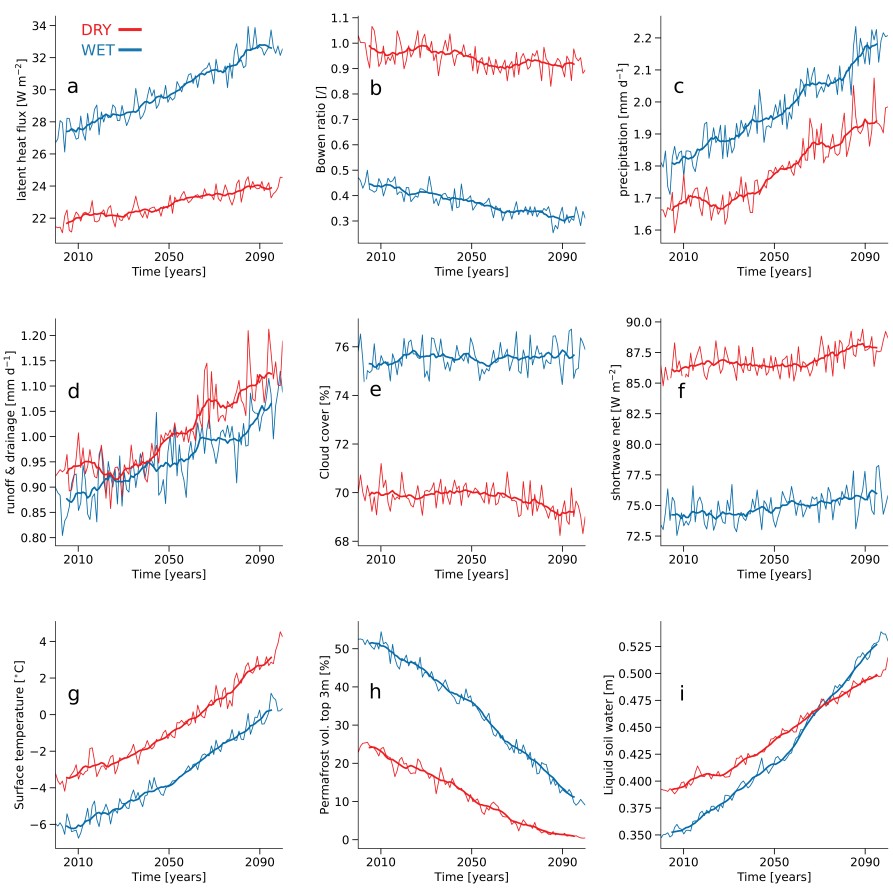

**Figure 6. Effects of soil hydrological conditions on near-surface climate:**
a) Latent heat flux in the northern permafrost regions in WET (blue) and DRY (red) MPI-ESM simulations for SSP5-8.5. Thin lines show the annual mean, averaged over the northern permafrost regions (note that that grid cells covered by glaciers were excluded), while thick lines give the 10-year running mean. b) Same as a) but showing the Bowen ratio, c) precipitation, d) surface runoff and drainage, e) accumulated cloud cover, f) solar radiation absorbed at the surface, g) surface temperatures, h) near-surface (top 3 m of the soil) permafrost volume and i) liquid soil water content.





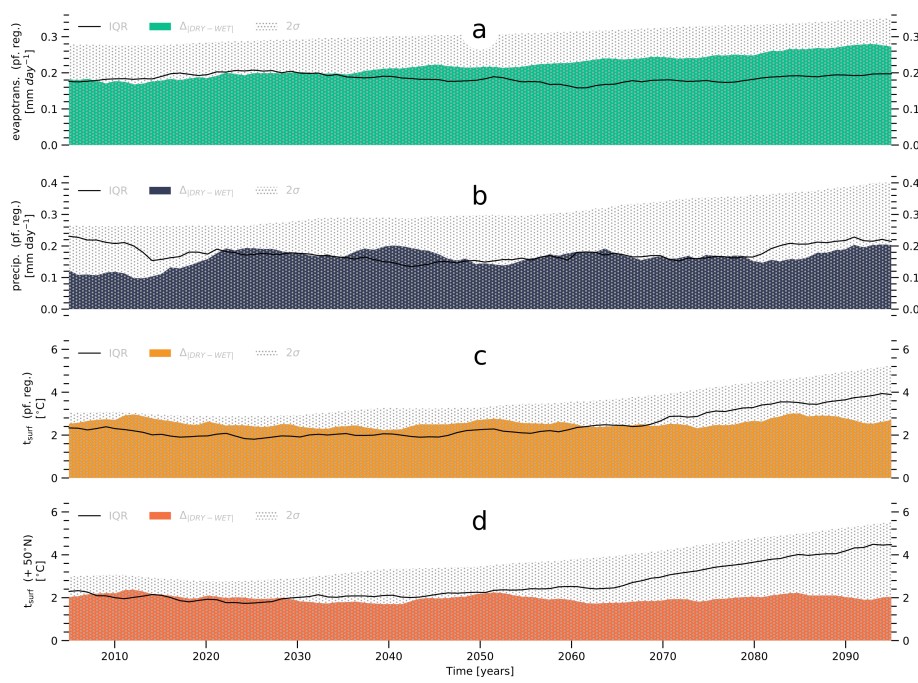

**Figure 7. Comparison to CMIP6 ensemble:**

a) Differences in evapotranspiration between the WET and the DRY setup in permafrost regions ($\Delta^{evap}_{|DRY-WET|}$). Black line gives the interquartile range (IQR$^{evap}$) — that is the difference between the 75$^{\text{th}}$ and the 25$^{\text{th}}$ percentile — of the CMIP6 ensemble, while the dotted area provides 2 × the ensemble standard deviation ($2\sigma^{evap}$). b) same as a but for precipitation ($\Delta^{pr}_{|DRY-WET|}$, IQR$^{pr}$, $2\sigma^{pr}$), c) surface temperatures in permafrost grid cells ($\Delta^{ts}_{|DRY-WET|}$, IQR$^{ts}$, $2\sigma^{ts}$) and d) surface temperatures (land and ocean) north of 50°N ($\Delta^{ts;+50N}_{|DRY-WET|}$, IQR$^{ts;+50N}$, $2\sigma^{ts;+50N}$).





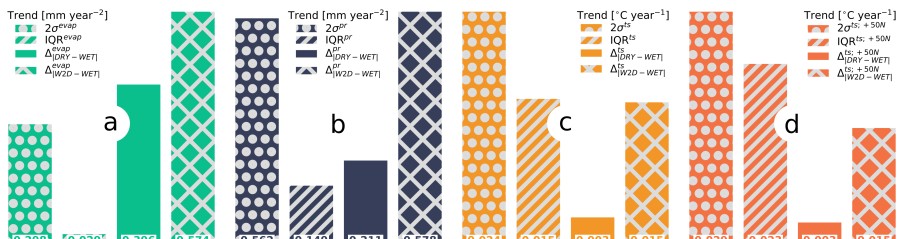

**Figure 8. 21ˢᵗ-century trends:**
a) Trends in evapotranspiration during the 21ˢᵗ century: $2 \times$ the CMIP6 ensemble standard deviation, the interquartile range of the CMIP6 ensemble, differences between WET and DRY and differences between W2D and WET — averaged over the northern permafrost regions. b) Same as a) but for precipitation and c) surface temperatures. d) Same as c) but for surface temperatures averaged over the region (land and ocean) north of 50°N.





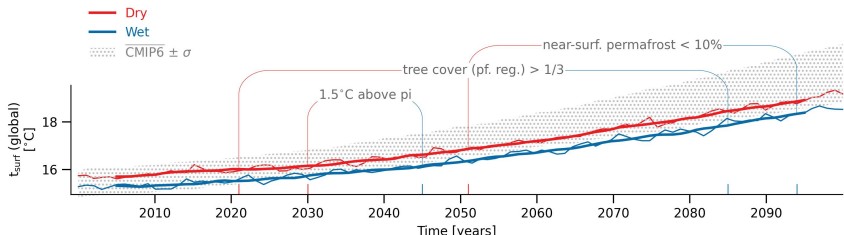

**Figure 9. Effect on global climate:**

Simulated global mean surface temperatures for the DRY (red), the WET (blue) setup and range of the CMIP6 ensemble mean ± one standard deviation. Shown are the points in time at which the tree cover in permafrost-affected regions exceeds on third of the surface area, the simulated global mean surface temperature reaches 1.5°C above pre-industrial levels (with the latter being based on the CMIP6 ensemble mean temperature) and the near-surface permafrost volume decreases below 10 %.

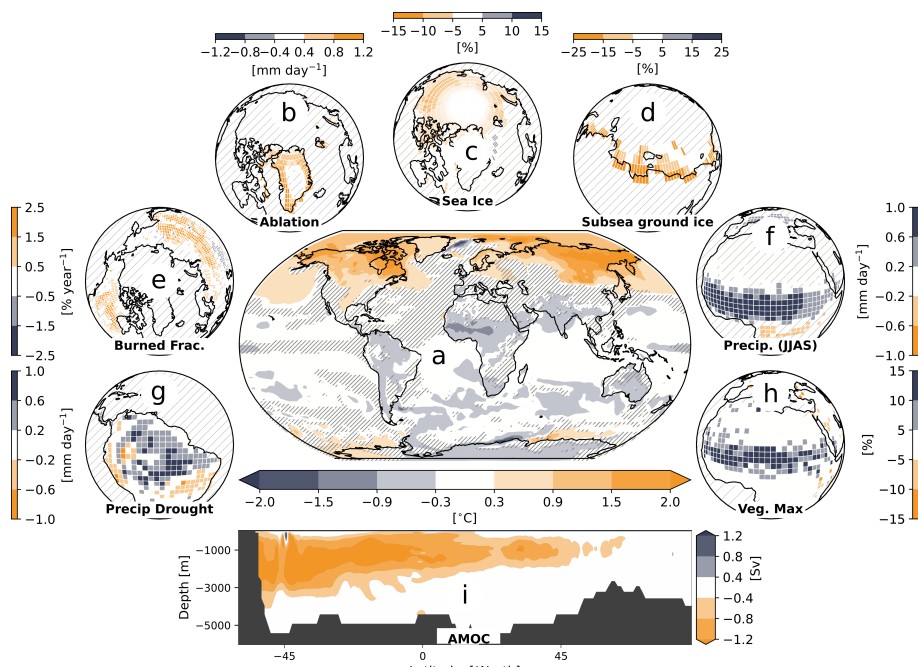

**Figure 10. State of tipping elements and other core climate elements at 1.5°C above the pre-industrial mean:**
a) Differences in the annual mean surface temperatures between the DRY and the WET setup for a global mean surface temperature of 1.5°C above the pre-industrial CMIP6 ensemble mean. Shown is the difference between simulations in which the atmospheric greenhouse gas concentrations were modified to stabilize the climate for a 50 year period. b) Same as a) but for glacier ablation rates, c) annual average Arctic sea-ice concentration and d) relative difference in subsea permafrost volume within the top 10 m of the subsurface. Shown is the Laptev Sea and the East Siberian Sea. Subfigure d is not based on simulations with the MPI-ESM but on simulations with an adapted JSBACH model that represents the permafrost dynamics on the Arctic shelves using the bottom temperatures from DRY and WET. A detailed description of this model version is given in Wilkenskjeld et al. (2021). e) Same as a) but showing the (annually) burned area in boreal regions, f) precipitation during west African monsoon, g) minimum 3-year mean precipitation in the Amazon basin, h) vegetation cover in the region of the west African monsoon and i) strength of the Atlantic Meridional Overturning Circulation. Shown is the 50-year mean. Areas where differences are not significant (p-value > 0.05) are hatched. Note that subfigure g) shows the minimum 3-year mean precipitation over land during the 50 year period without test of significance.





**Table 1.** Overview over the DRY and WET JSBACH-setup

| Sim. | $W_{scool}$ mobile | $F_{drain,l}$ used | $W^*_{MAX,rz}$ | $W^*_{MAX,top}$ | $F_{ARNO}$ [/] | $P_{lag}$ [d] | $d_{rz,min}$ [m] |
|---|---|---|---|---|---|---|---|
| DRY | True | False | $W_{MAX,rz} + \Delta\Phi_{org} - W_{ice,rz}$ | $W_{MAX,top} + \Delta\Phi_{org} - W_{ice,top}$ | 0.8 | 100 | 0.1 |
| WET | False | True | $W_{MAX,rz} - W_{ice,rz}$ | $W_{MAX,top} - W_{ice,top}$ | 1.0 | 150 | 0.3 |



**Table 2.** CMIP6 participants included in analysis

| Models | Institution |
|---|---|
| ACCESS-ESM1.5; ACCESS-CM2 (Bi et al., 2020; Ziehn et al., 2020) | Commonwealth Scientific and Industrial Research Organisation; Australian Research Council Centre of Excellence for Climate System Science (Australia) |
| AWI (Semmler et al., 2020) | Alfred Wegener Institute, Helmholtz Centre for Polar and Marine Research (Germany) |
| BCC-CSM2-MR (Wu et al., 2019) | Beijing Climate Center (China) |
| CAMS-CSM1.0 (Rong et al., 2018) | Chinese Academy of Meteorological Sciences (China) |
| CanESM5-CanOE; CanESM5 (Swart et al., 2019) | Canadian Centre for Climate Modelling and Analysis (Canada) |
| CESM2-CAM6; CESM2-WACCM6 (Danabasoglu et al., 2020) | National Center for Atmospheric Research (USA) |
| CIESM (Lin et al., 2020) | Department of Earth System Science, Tsinghua University (China) |
| CMCC-ESM2; CMCC-CM2-SR5 (Cherchi et al., 2019) | Centro Euro-Mediterraneo sui Cambiamenti Climatici (Italy) |
| CNRM-CM6.1-HR; CNRM-CM6.1; CNRM-ESM2.1 (Roehrig et al., 2020; Michou et al., 2020; Voldoire et al., 2019) | Centre National de Recherches Meteorologiques; Centre Europeen de Recherche et de Formation Avancee en Calcul Scientifique (France) |
| E3SM-1.1 (Golaz et al., 2019; Burrows et al., 2020) | Department of Energy (USA) |
| EC-Earth3; EC-Earth3-Veg0; EC-Earth3-CC; EC-Earth3-Veg-LR (Döscher et al., 2021) | EC-Earth consortium, Rossby Center, Swedish Meteorological and Hydrological Institute (Sweden) |
| FGOALS-f3-L; FGOALS-g3 (He et al., 2019; Li et al., 2019; Bao and Li, 2020) | Institute of Atmospheric Physics (China) |
| FIO-ESM-2.0 (Bao et al., 2020) | First Institute of Oceanography (China) |
| GFDL-CM4; GFDL-ESM4 (Held et al., 2019; Dunne et al., 2020) | National Oceanic and Atmospheric Administration, Geophysical Fluid Dynamics Laboratory (USA) |
| GISS-E2.1-G (Kelley et al., 2020; Miller et al., 2021) | Goddard Institute for Space Studies (USA) |
| IITM-ESM (Swapna et al., 2018) | Centre for Climate Change Research, Indian Institute of Tropical Meteorology (India) |
| INM-CM5.0; INM-CM4.8 (Volodin et al., 2017, 2018) | Institute for Numerical Mathematic (Russia) |
| IPSL-CM6A-LR (Boucher et al., 2020; Hourdin et al., 2020; Lurton et al., 2020) | Institut Pierre-Simon Laplace (France) |
| K-ACE-1-0-G (Lee et al., 2019) | National Institute of Meteorological Sciences, Korea Meteorological Administration (South Korea) |
| KIOST-ESM (Pak et al., 2021) | Korea Institute of Ocean Science and Technology (Korea) |
| MIROC6; MIROC-ES2L (Tatebe et al., 2019; Hajima et al., 2020) | Japan Agency for Marine-Earth Science and Technology; National Institute for Environmental Studies; Atmosphere and Ocean Research Institute, The University of Tokyo; RIKEN Center for Computational Science (Japan) |
| MPI-ESM1.2-LR; MPI-ESM1.2-HR (Mauritsen et al., 2019; Müller et al., 2018) | Max Planck Institute for Meteorology; Deutsches Klimarechenzentrum; Deutscher Wetterdienst (Germany) |
| MRI-ESM2.0 (Yukimoto et al., 2019) | Meteorological Research Institute (Japan) |
| NESM3 (Cao et al., 2018) | Nanjing University of Information Science and Technology (China) |
| NorESM2-LM; NorESM2-MM (Seland et al., 2020; Tjiputra et al., 2020; Counillon et al., 2016) | Norwegian Climate Center (Norway) |
| TaiESM (Lee et al., 2020) | Research Center for Environmental Changes, Academia Sinica (Taiwan) |
| UKESM1.0-LL; HadGEM3-GC31-LL; HadGEM3-GC31-MM (Sellar et al., 2019; Kuhlbrodt et al., 2018; Williams et al., 2018) | Met Office Hadley Center; Natural Environment Research Council (UK) |