# Peer review of "Representation of soil hydrology in permafrost regions may explain large part of inter-model spread in simulated Arctic and subarctic climate"

_The Cryosphere, 2022_

## Referee Comment (RC1)

Review of "Representation of soil hydrology in permafrost regions may explain large part of inter-model spread in simulated Arctic and subarctic Climate" by Philipp de Vrese et al. submitted to *The Cryosphere*:

The article by Philipp de Vrese et al. presents a set of coupled simulations using the MPI Earth system model in which parameterizations of the soil hydrology within the permafrost region have been varied. The effects of these variations on land-atmosphere fluxes in the permafrost region are investigated and also effects on the global climate are discussed. The authors compare the variation among the presented simulations to variations among the projections of CMIP6 models and interpret that a the inter-model spread in key climate variables of the high latitudes can be attributed to differences in the models' parameterizations of soil hydrology in areas affected by permafrost.

The question of whether the high latitudes become wetter or drier under future climate change is unanswered and at the same time highly relevant for Arctic climate feedbacks such as the permafrost carbon feedback. While the present article does not answer this question, it does provide new insights which help to identify sources of uncertainty of climate model predictions and points to assumptions and parameterizations in current model which require more attention and careful consideration by climate modellers. Thus, the article is very relevant within the scope of TC and strongly builds on works which have previously been published in this journal.

The methodology of the study is explained well and adequate to address the research objectives. The results are described clearly and mostly explained in an understandable way. The overall presentation quality is good, but some figures should be revised (see comments below). Less clear to me was the distinction between Results, Discussion, and Conclusions which appear to be mixed to certain degree. Here, the paper would benefit from restructuring and rebalancing of the presented contents (see comments below).

Overall, the paper addresses a relevant topic, the results are substantial and presented well. In principle I support its publication in TC. Before the paper can be accepted, however, the following comments need to be taken into consideration by the authors:

**General comments**

- A main point of the study is, that different parameterizations of the soil hydrology in permafrost regions could explain inter-model spread in climate variables. The authors explain in detail how the soil hydrology is parameterized in MPI-ESM and how they modified it to come up with two contrasting but plausible setups (WET and DRY). However, the authors do not establish a connection to the respective parameterizations in other state-of-the-art land surface schemes. In order to not only quantitatively (e.g. Fig. 6), but also qualitatively substantiate the explanation of inter-model spread due to soil hydrology parameterizations, it would be very helpful if the authors would (i) introduce briefly typical assumptions and

parameterizations of the soil hydrology in the current generation of LSMs (in the Introduction; an overview Table could be helpful for this), and (ii) discuss the findings using the two setups of MPI-ESM in relation to typical setups of other land surface schemes in CMIP6 models (in the Discussion section).

- The description of the modifications to the model (section 2.1) and its setups (section 2.2) would benefit a lot from a schematic illustration of the soil hydrology processes and their parameterizations. For example, such a figure could illustrate the differences between runoff, infiltration, drainage, and how these fluxes are affected by the state of the surface and subsurface. Such a schematic figure might also be used to illustrate the different assumptions made for the WET and DRY setups.

- I am wondering how useful the approach of applying the modified parameterizations only to a pre-defined sub-region of the land model domain (the present-day permafrost extent based on Hugelius et al. 2014) is. The biogeophysics and biogeochemistry follows universal laws such that it should be possible to come up with general parameterizations which can be applied over the entire land domain and not only in an pre-defined region whose borders are set "arbitrarily". While I see that the approach is sufficient to address the research questions of this study, it brings a lot of shortcomings with it. For example, it introduces additional inter-model uncertainties due to the definition of the region which makes inter-model comparisons hard. In addition, the definition of the region is based on a present-day snapshot of permafrost occurrence which in turn is dynamic and would be very different under past or future climate conditions. I would appreciate a discussion of this issue and if the authors could line out a way towards a more sustainable modelling approach for modified permafrost representations.

- The authors conduct and present a set of simulations for which the climate reaches stabilization at an GMST increase of 1.5° the results of which I find very insightful (Fig. 10). In fact, these are two coupled simulations for the WET and DRY setup, respectively, which are using overshoot scenarios in addition to the two simulations under SSP5-8.5. Therefore, I find it rather surprising that the results of these simulations are presented within the "Conclusions" section (which reads more like a Discussion section). It would help the overall structure and accessibility of the article, if (i) an appropriate research objective would be provided in the Introduction which motivates these simulations, (ii) the simulation results would be presented in a dedicated Results section which addresses that research question.

- I appreciate the effort of the authors to also conduct numerical experiments with ICON-ESM to substantiate their findings. However, I am not convinced that the way this is currently presented in the article is helpful for the readers. The ICON-ESM simulations are described in a dedicated Methods section 2.4, but not mentioned thereafter, neither in the results, nor in the discussion or conclusions. If it is a specific objective of the authors to substantiate their findings also with the simulations of ICON-ESM, then these should also be mentioned in the Results and Discussion sections. Otherwise, the authors might consider to omit presenting the ICON-ESM simulations at all in the present study.

- The final section which is labelled "Conclusions" provides mostly a discussion of the results, and in addition introduces further model results obtained using overshoot scenarios. However, the section does not present any concrete conclusions of the present work in a comprehensive and condensed way. I strongly suggest to carefully revise the structure and contents presented in this section. Connecting to my comment regarding the 1.5° simulations above, I suggest that these results should be presented in a dedicated subsection of the "Results" section. All results (land–atmosphere interactions, comparison to the CMIP6 spread, (optionally the ICON-ESM comparison), and the comparison of the overshoot runs) could then be discussed in a dedicated Discussion section building on the current "Conclusions" section. An additional section which provides the study's conclusions in a condensed way would further improve the article's accessibility and impact.

- The Discussion of the results does not sufficiently consider the parameterization of soil hydrology in other land surface schemes (see first comment above). An additional paragraph in which the present work with MPI-ESM is related to other model configurations could help to substantiate the findings regarding the inter-model spread also qualitatively.

**Specific comments**

- l.109ff: The authors state that the model representation of soil physics is largely based on the modifications by Ekici et al. and explain the differences. However, it was not clear to me how the model version used relates to that described in de Vrese et al. (2021) where several modifications to the soil physics and biogeochemistry have been described. Maybe I overlooked this, but are these modifications also included in the model version used for the present study?

- 1.124f: "but includes their influence on the hydrological soil properties." Does this refer to the modifications described in sections 2.1.2.-2.1.4 or are there further ways in which the organic matter influences the hydrological soil properties (e.g. hydraulic conductivities). If the latter is the case, all equations should be provided in the main text or the appendix. Otherwise it should be made clear, that this statement refers to the modifications described later on.

- Section 2.1.1. mentions a lot of issues and problems which faced by soil hydrology parameterizations, in particular in JSBACH. However, it is not always clear which of these issues are being addressed in the present study and which are not.

- l.158: It is not clear how infiltration, especially into frozen ground, is controlled in the current version. The authors only state that it is "exclusively controlled by the saturation of the near-surface soil layers ...", but they do not provide any equations or references to other work where this is described.

- 1.225: The authors state that the field capacity constitutes the upper limit of the soil water

content in JSBACH, which I found confusing. I would have expected that it is possible to have soil water contents exceeding the field capacity, for example under saturated conditions. For example, in 1.248 it is written that "only the water that exceeds the layer's field capacity is added to the drainge flux" which in my view contradicts the formulation in 1.225. Am I missing something here? A clarification would be appreciated.

- 1.292: It is not clear to which version "JSBACH's original formulation" refers. It is stated that the original formulation would prohibit infiltration at sub-zero temperatures, but I understood that this had been only introduced into the model by Ekici et al. Also in light of my comment below on using the term "reference model", it might be helpful to clearly designate the different JSBAHC model versions/configurations referred to in this paper.

- l.314ff: It is not clear how the subsurface parameters have been set, since the deep model configuration deviates from the standard one. I suspect that the same datasets as in González-Rouco et al., 2021 and Steinert et al. 2021a have been used? This should be clarified in order to allow reproducibility.

- l.414: Why not stating the proportion of simulations contained in two standard deviations (~95%) if these are used?

- l.422/Fig. 5: Here and later the authors often refer to "trends" which are shown in Fig. 5. How have these trends been estimated? (linear regression?, which time period?, ...). Please specify this, ideally already in the Methods section.

- l.433ff: I do not understand the argumentation regarding the effect of excess ice. I would expect that, in a first-order approximation, the inclusion of excess ice and associated ground subsidence would result in wetter conditions if thaw depths increase, since the ground subsidence counteracts the possibility of deeper infiltration (see Fig. 8 in Andresen et al. (2020)).

- The final paragraph of the results section (1.538ff) introduces an additional simulation setup (W2D). Admittedly, I had a hard time understanding this paragraph. This part would benefit from a bit more detailed explanations of the motivation for and the interpretation of the additional simulations. It might also help to replace some of the rather technical language which uses symbols as parts of the sentences by verbal descriptions.

- l.589: It might be interesting to explicitly state the high latitudes become warmer under the DRY relative to the WET setup while the tropics and mid latitudes are colder under the WET setup relative to the dry setup.

- 1.603f: Please provide references for the statement regarding the effect of sea ice.

- l.615: It should be stated in the text that the subsea permafrost was diagnosed using a different model configuration.

- 1.624: It is not clear what "negatively affected" means in this context.

- l.662: The work of Aas et al. (2019) might also be relevant in this context.

- l.664f: It would be interesting to discuss whether ICON-L/JSBACH4 provide this functionality.

- To my opinion, Figure 3 is not crucial for understanding the methodology and should go into the appendix or supplement.

- The caption of Fig. 4 refers to a "reference model" abbreviated REF, which is, however, not introduced at any point in the main text. It is not clear whether this is the standard configuration of MPI-ESM/JSBACH3.2 or whether it includes the modifications by Ekici et al. Please clarify.

- Fig. 5/6: While Fig. 5 shows the trends in total water content, Fig. 6i shows the aggregated changes in liquid water content. However, it would also be interesting to see how the absolute values of total water content evolve during the 21st century and how the two setups compare to each other to get the full picture.

- Fig. 6: The differences in permafrost volume (panel h) are quite huge among the two setups. It would be interesting to also show the simulated permafrost areal coverage and mention how it compares to estimates for the present day coverage. Are other estimates contained within the spread between the WET and DRY scenarios?

- Fig. 7: I do not understand why this figures uses a mix of "area" and "line" plot features. It would make more sense if the IQR and 2sigma would be indicated as areas, and the Delta as lines. But showing all quantities as a line plot should also work.

- Fig. 7/8: To be consistent and to provide the full picture, you might consider to include the plots and values for the region north of 50°N also for precipitation and evapotranspiration.

- Fig. 9: The scenario (SSP5-8.5) should be mentioned in the caption.

- Tab. 2 should go into the supplement.

**Technical corrections**

- 1.29: Should be "as contained in the Earth's atmosphere."

- l.137: "be" missing in "could derived from observations."

- l.150: "see below" --> Please be more specific and refer to specific sections.

- l.157: typo: "soil" instead of "seoil"

- Fig. 5: I don't see why you write "EVAP.", "PR." and "H20" instead of "Evaporation", "Precipitation" and "Total soil water" in the heading.

**References**

Aas, K. S., Martin, L., Nitzbon, J., Langer, M., Boike, J., Lee, H., Berntsen, T. K., & Westermann, S. (2019). Thaw processes in ice-rich permafrost landscapes represented with laterally coupled tiles in a land surface model. The Cryosphere, 13(2), 591–609. https://doi.org/10.5194/tc-13-591-2019

All other references are contained in the manuscript.

---

## Author Comment (AC1)

**Response to comments of reviewer 1**

Before the point-by-point address of the reviewer's comments, we would like to thank Dr. Nitzbon for his very detailed review of our manuscript. We apologize in advance for not being able to act on all of the suggestions – as we explaine in more detail below, it was beyond our abilities to establish a clear connection to the parametrizations used in other LSMs – nonetheless, we hope that we managed to make good use of all his comments.

*Please note that, in the following point by point address, we repeat the Dr. Nitzbon's comments in red letters while **our response** is given in black letters. All the post-reply updates are indicated in blue.*

Review of „Representation of soil hydrology in permafrost regions may explain large part of inter-model spread in simulated Arctic and subarctic Climate" by Philipp de Vrese et al. submitted to The Cryosphere:

The article by Philipp de Vrese et al. presents a set of coupled simulations using the MPI Earth system model in which parameterizations of the soil hydrology within the permafrost region have been varied. The effects of these variations on land-atmosphere fluxes in the permafrost region are investigated and also effects on the global climate are discussed. The authors compare the variation among the presented simulations to variations among the projections of CMIP6 models and interpret that a the inter-model spread in key climate variables of the high latitudes can be attributed to differences in the models' parameterizations of soil hydrology in areas affected by permafrost.

The question of whether the high latitudes become wetter or drier under future climate change is unanswered and at the same time highly relevant for Arctic climate feedbacks such as the permafrost carbon feedback. While the present article does not answer this question, it does provide new insights which help to identify sources of uncertainty of climate model predictions and points to assumptions and parameterizations in current model which require more attention and careful consideration by climate modellers. Thus, the article is very relevant within the scope of TC and strongly builds on works which have previously been published in this journal.

The methodology of the study is explained well and adequate to address the research objectives. The results are described clearly and mostly explained in an understandable way. The overall presentation quality is good, but some figures should be revised (see comments below). Less clear to me was the distinction between Results, Discussion, and Conclusions which appear to be mixed to certain degree. Here, the paper would benefit from restructuring and rebalancing of the presented contents

(see comments below).

Overall, the paper addresses a relevant topic, the results are substantial and presented well. In principle I support its publication in TC. Before the paper can be accepted, however, the following comments need to be taken into consideration by the authors:

**General comments**

- A main point of the study is, that different parameterizations of the soil hydrology in permafrost regions could explain inter-model spread in climate variables. The authors explain in detail how the soil hydrology is parameterized in MPI-ESM and how they modified it to come up with two contrasting but plausible setups (WET and DRY). However, the authors do not establish a connection to the respective parameterizations in other state-of-the-art land surface schemes. In order to not only quantitatively (e.g. Fig. 6), but also qualitatively substantiate the explanation of inter-model spread due to soil hydrology parameterizations, it would be very helpful if the authors would (i) introduce briefly typical assumptions and parameterizations of the soil hydrology in the current generation of LSMs (in the Introduction; an overview Table could be helpful for this), and (ii) discuss the findings using the two setups of MPI-ESM in relation to typical setups of other land surface schemes in CMIP6 models (in the Discussion section).

Indeed, when we designed our setups we did not only try to match the spread of the PCN-MIP ensemble, but we also relied on the suggestions of Andresen et al. (2020) as to what may constitute the most important differences between models. Here, they proposed the representation of evapotranspiration, soil organic matter, the water table – which in some models is used to estimate infiltration rates – and the vertical movement of water through the soil as key factors of the models' soil hydrology schemes (see the discussion on "Uncertainty in soil moisture and hydrology simulations" in Andresen et al. (2020)). However, even their detailed analysis – that relied on simulations performed in the setting of a dedicated model intercomparison project, in which the LSMs are driven by a similar climate forcing – couldn't fully detail the range of assumptions in present-day models, resolve how specific implementations affect the hydrological processes or quantify how (relatively) important the individual factors are.

One reason for this is that many models use similar approaches to represent key processes but employ slightly different formulations, the details of which are often not even described in the model documentation but only become apparent in the model's source code. A good example for this problem is the supercooled water. Here, the information if and how a model parameterizes supercooling is readily available, but the question whether this water is available for plants can, in many cases, not be answered from the available model documentation. Another important reason is that some effects can not be attributed to one specific formulation but to its interactions with other parts of the code. Here, our own model provides a very good example, as it took us quite some time to work out that the strong warming that the implementation of Ekici et al. (2014)

caused, was in large parts due to the way the new 5-layer snow scheme affected the infiltration rates. As we also describe in the method section, infiltration was not permitted when the temperature of the upper most layer was below zero °C, which was not a problem as long the uppermost layer still represented the melting snow (which was at exactly zero °C during snowmelt). However, with the new snow module placing the the snow on top of the soil column, the temperature of the uppermost layer was below zero °C during the snowmelt. This resulted in the entirety of the meltwater running off at the surface, leading to extremely dry soils in spring and summer.

Thus, in all honesty, we do not know whether our setups describe the range of assumptions in current land surface models well or only capture the bulk uncertainty. For these reasons it is exceedingly difficult to relate our findings to the setups of other land surface schemes and it is extremely uncertain that the adaptations we made to our model would have the same effect in a different LSM. Thus, we are very happy to include a table showing the key features of the soil hydrology in the PCN-MIP participants (adapted from Andresen et al. (2020)), and include a very brief discussion of the above issues in the supplementary materials (see below) and in the discussion section (see comment on the Conclusions below). However, we can not qualitatively assess how specific soil hydrology parameterizations affect the inter-model spread.

**S1 Soil hydrology representation in PCN-MIP participants**

*"Many factors that are especially relevant for permafrost-affected areas are treated very differently by current-generation land surface models. Here, one prominent example is the process of supercooling which is included in roughly half of the models participating in the PCN-MIP (Andresen et al., 2020). In clay rich soils a substantial fraction of water may remain liquid at sub-zero temperatures and potentially move through and drain from the soil even if the model considers the effect of soil ice on the hydraulic conductivity – which many models do using a power-law-based ice impedance factor. Another, important factor is the treatment of soil organic matter, which is also explicitly included in about half of the PCN-MIP participants. Here, the importance of a high soil organic matter concentration is often attributed to its impact on the soil thermal properties, as the high porosity can provide a strong insulation at the surface of dry soils. However, soil organic matter also affects the hydrological properties, increasing conductivity, hence, the percolation into deeper layers and drainage. This affects the water available for transpiration and evaporation, with a strong impact on the surface energy budget.*

*"At the same time, most present-day land surface models use similar formulations to represent some of the key processes of the terrestrial hydrology (Andresen et al., 2020; Telteu et al., 2021). For example, the majority of models employs a 1-D formulation based on the Richards equations with Clapp–Hornberger or van Genuchten functions to describe the vertical movement of water through the soil, while surface runoff and infiltration are most often described using a saturation-excess based formulation. But while these approaches appear similar, they may yield diverging results, depending on the specifics of their implementation. For example, the infiltration in a saturation-excess based formulation depends on the way the model determines the surbgrid scale distribution of the water table or, if the model instead uses the soil field capacity or porosity as an*

*upper limit for infiltration, on the assumption up to which soil layer water can infiltrate within one timestep. The same is true for the representation of evapotranspiration as most models account for the same sources but employ different formulations to describe the respective fluxes.*

*"It is exceedingly difficult to quantify how a given parametrization affects the modelling outcome as it is very often the interactions between processes that determine the behaviour of a given model. This is particularly problematic as differences in the simulated soil hydrology may also originate from differences in the treatment of the soil thermal dynamics – as these determine the state of water in the soil – as well as from differences in the general model setup, e.g. vertical resolution and depth of the soil column, and the representation of vegetation. Thus, our JSBACH setups merely aim at capturing the bulk effects of the uncertainty included in the range of established soil hydrology representations without trying to connect them to specific formulations employed by present-day land surface models. An overview of the hydrology characteristics of the PCN-MIP participants is provided in table ST1, while a more detailed discussion can be found in Andresen et al. (2020)."*

| Model | Evapotranspiration | Root water uptake | Infiltration | Water table | Soil water storage and transmission | Ground-water dynamics | Soil-ice impact | Snow | Super-cooling | Organic matter |
|---|---|---|---|---|---|---|---|---|---|---|
| CLM4.5 | Sum of canopy evaporation, transpiration and soil evaporation | Macroscopic approach | Saturation-excess runoff Fsat=f(zwt) | perched water table possible if ice layer present[3] | Richards equation (Clapp–Hornberger functions) | Base flow from TOP-MODEL concepts, unconfined aquifer[3] | Impacts hydrologic properties through power-law ice impedance[4] | Multilayer dynamic (5 max) | Yes | Yes |
| CoLM | Biosphere– atmosphere transfer scheme and Philip's[5] | Macroscopic approach | Saturation-excess runoff Fsat=f(zwt) | Simple TOP-MODEL | Richards equation (Clapp–Hornberger functions) | Base flow from TOP-MODEL | Impacts hydrologic properties through power-law ice impedance | Multilayer dynamic (5 max) | No | No |
| JULES | Sum of ET, soil evaporation and moisture storages (e.g., lakes, urban) minus surface resistance | Macroscopic approach | Saturation-excess runoff Fsat=f(zwt) or Fsat=f(θ) | TOPMODEL or probability distribution model | Richards equation (Clapp– Hornberger and van Genuchten functions) | Base flow from TOP-MODEL | Hydraulic conductivity and suction determined by unfrozen-water content (Brooks and Corey functions) | Multilayer dynamic (3 max) | Yes | No |
| ORCHIDEE-IPSL | Sum of bare soil, interception loss and plant transpiration for different vegetation PFTs in grid cell | Macroscopic approach, water uptake different among cell vegetation PFTs[6] | Saturation-excess runoff Fsat=f(θ) | TOPMODEL | Richards equation (van Genuchten functions) | None | "Drying = freezing" approximation[7] | Multilayer dynamic (7 max) | Yes | Yes |
| LPJGUESS | Sum of interception loss, plant transpiration and evaporation from soil[8] | Fractional water uptake from different soil layers according to prescribed root distribution[9,10] | Depends on soil moisture and layer thickness, declines exponentially with soil moisture | Uniform and only for wetland grid cell[9,10] | Analog to Darcy's law, percolation rate depends on soil texture conductivity and soil wetness[11] | Base flow is based on the exponential function to estimate percolation rate | Impacts hydrologic properties through power-law ice impedance | Multilayer dynamic (3 max) | No | No |
| SIBCASA | Sum of ground evaporation, surface dew,canopy ET and canopy dew[12] | Macroscopic approach | Infiltration approach in non- saturated porous media described by Darcy's law | perched water table possible if ice layer present[3] | Richards equation(Clapp-Hornberger functions) | Base flow from TOP-MODEL concepts, unconfined aquifer[3] | Impacts hydrologic properties through power-law ice impedance | Multilayer dynamic (5 max) | Yes | Yes |
| TEM-604 | Jensen–Haise potential ET[13]; actual ET is calculated based onPET, water availability and leaf mass | Based on the proportion of actual ET to potential ET | Field capacity ex-cess runoff[14] | None | One-layer bucket | None | None | Multilayer dynamic (9 max) | No | No |
| UWVIC | Sum of canopy interception, vegetation transpiration and soil evaporation[5] | Based on reference ET and soil wilting point | Saturation-excess runoff Fsat=f(θ) | Microtopography | From infiltration rate and infiltration shape parameter[15], no lateral flow between model grids | Base flow from Arno model conceptualization[16] | Impacts hydrologic properties through power-law ice impedance | Bulk-layer dynamic (2 max) | Yes | Yes |
| JSBACH (standard) | Sum of transpiration, bare soil, canopy and snow evaporation | Macroscopic approach | ARNO - rainfall - runoff model[17] | None | Richards equation (van Genuchten function) | None | None | 1 layer dynamic | No | No |
| JSBACH (WET) | Sum of transpiration, bare soil, canopy and snow evaporation | Macroscopic approach | ARNO model combined with WEED scheme[18] | perched water table possible on top of permafrost | Richards equation (van Genuchten function) | None | Impacts hydrologic properties through power-law ice impedance (with no movement of soil water about 50% sturation of pore space with ice) | Multilayer dynamic (5 max) | Yes | Yes |
| JSBACH (DRY) | Sum of transpiration, bare soil, canopy and snow evaporation | Macroscopic approach | ARNO model combined with WEED scheme[18] | perched water table possible on top of permafrost | Richards equation(van Genuchten functions) | None | Impacts hydrologic properties through power-law ice impedance | Multilayer dynamic (5 max) | Yes | Yes |

**Supplementary Table ST1.** Hydrology characteristics of land surface models participating in the PCN-MIP. Note that the table was adapted from Andresen et al. (2020)[1] and that JSBACH was initially not part of the PCN-MIP.

- The description of the modifications to the model (section 2.1) and its setups (section 2.2) would benefit a lot from a schematic illustration of the soil hydrology processes and their parameterizations. For example, such a figure could illustrate the differences between runoff, infiltration, drainage, and how these fluxes are affected by the state of the surface and subsurface. Such a schematic figure might also be used to illustrate the different assumptions made for the WET and DRY setups.

We agree with Dr. Nitzbon that it could be helpful to include a graphical summary of the two JSBACH setups. However, as stated above, the adaptations that we used to create the setups are very model-specific and may not lead to the same outcomes when used in other land surface models. Therefore, we would prefer to not place too strong an emphasis of the details of the implementation, i.e. by including a schematic illustration of JSBACH's soil hydrology. Instead we would rather show – qualitatively – the effect that the combination of parameter choices has on the land atmosphere interactions.

[Figure]

**Figure 3. JSBACH setups**

Qualitative comparison between the hydrological fluxes in the DRY and WET JSBACH setups. Shown are bare-soil evaporation (yellow), evaporation from wetlands (dark blue), transpiration (green), infiltration (white), surface runoff (grey) and drainage (light blue), with thick arrows indicating large fluxes and thin arrows small fluxes. The resistances symbols (red in case of the DRY and blue for WET setup) indicate whether the parameter settings in a setup facilitate a certain process – indicated by a small resistance symbol – or impede it – indicated by a large resistance symbol. Here, the resistances with respect to bare soil evaporation and transpiration depends on the parametrization of the maximum water holding capacity of the uppermost soil layer and of the rootzone, respectively. The resistance with respect to evaporation from wetlands can be modified via the maximum retention time, which determines the surface water storage and the fractional wetland cover. With respect to infiltration and surface runoff the resistance depends on the scaling factor $F_{ARNO}$ which allows to reduce the flux from the surface water storage to the ARNO scheme. The resistance with respect to drainage depends on the assumed mobility of supercooled water as well as on the ice-impedance factor.

- I am wondering how useful the approach of applying the modified parameterizations only to a pre-defined sub-region of the land model domain (the present-day permafrost extent based on Hugelius et al. 2014) is. The biogeophysics and biogeochemistry follows universal laws such that it should be possible to come up with general parameterizations which can be applied over the entire land domain and not only in an pre-defined region whose borders are set "arbitrarily". While I see that the approach is sufficient to address the research questions of this study, it brings a lot of shortcomings with it. For example, it introduces additional inter-model uncertainties due to the definition of the region which makes inter-model comparisons hard.

In addition, the definition of the region is based on a present-day snapshot of permafrost occurrence which in turn is dynamic and would be very different under past or future climate conditions. I would appreciate a discussion of this issue and if the authors could line out a way towards a more sustainable modelling approach for modified permafrost representations.

The fact that biogeophysics and biogeochemistry follow universal laws was essentially the "problem" for our study, which targets the impacts of processes in a specific region. It is a combination of factors that make the soil hydrology in permafrost affected areas special, but very few of those factors are actually unique to the region. For example, high concentrations of soil organic matter play an important role in the land-atmosphere interactions in the high latitudes, but in simulations with a global model it would be very hard to separate the respective effects from those orientating in, e.g., tropical peatlands. Similarly, the effects of soil ice are particularly important if this ice does not melt during summer, but the respective formulations act everywhere where soil water freezes during winter. Thus, we saw the application of a mask as the only viable (though admittedly not very elegant) solution to our problem of limiting the differences in setups to the permafrost regions. However, we don't see this as a general issue, as the need to limit effects to a specific region is nothing that is required for most simulations. For a general modelling approach it is perfectly fine if factors that are key in the permafrost hydrology also have effects in other regions. We certainly do not make this mask a part of our standard JSBACH model.

- The authors conduct and present a set of simulations for which the climate reaches stabilization at an GMST increase of 1.5° the results of which I find very insightful (Fig. 10). In fact, these are two coupled simulations for the WET and DRY setup, respectively, which are using overshoot scenarios in addition to the two simulations under SSP5-8.5. Therefore, I find it rather surprising that the results of these simulations are presented within the "Conclusions" section (which reads more like a Discussion section). It would help the overall structure and accessibility of the article, if (i) an appropriate research objective would be provided in the Introduction which motivates these simulations, (ii) the simulation results would be presented in a dedicated Results section which addresses that research question.

We agree that it makes perfect sense to present these findings in a dedicated result section. Hence, we introduced the sub-section (beginning in line 571 of the revised manuscript):

*"Relevance for global climate and the state of tipping elements"*

in which we present all the results for the climate stabilization at at 1.5°above pre-industrial levels. Furthermore, we changed the end of the introduction to (line 91 of the revised manuscript):

*"Section 2 details the changes to the model and the setup of the simulations, while section 3 discusses the pathways by which the uncertainty in the soil hydrology affects the Arctic and subarctic climate, compares the magnitude of the climate effects to the spread of the current CMIP model ensemble and investigates their relevance for the global scale with a particular focus on*

*their impact on a number of tipping elements."*

Please note that we also removed panel d from Figure (formerly) 7 and present it now in a separate figure to make the transition from the local (i.e. the terrestrial permafrost regions) to the global scale. However, as we mainly discuss the latter with respect to the latitudinal temperature gradient, we limit Figure 5 to the temperatures and do not show evapotranspiration and precipitation north of 50°N (line 571 of the revised manuscript).

*"Relevance for global climate and the state of tipping elements*

*While the above results show that the differences in the parametrization of the permafrost hydrology may not fully explain the spread of the CMIP6 ensemble – especially the latter's seasonality and 21ˢᵗ-century trend – the differences in simulated climate of the continental permafrost areas are substantial. This raises the question whether the respective effects are confined to this region or whether they are relevant for the climate on larger scales. Non-glaciated, permafrost-affected areas only cover about a third of the Arctic and subarctic zone. Yet averaged over the planet's surface north of 50°N the temperature differences between WET and DRY ($\Delta^{ts;+50N}_{|DRY-WET|}$) amount to about 2.0 °C (Fig. 10). This is notably less than $\Delta^{ts}_{|DRY-WET|}$, but still more than twice as much as would have been the case if the temperature effects were limited to the land areas affected by permafrost, indicating that the permafrost hydrology is indeed relevant for the entire region, including glaciers and the ocean.*

[Figure]

**Figure 10. Comparison to CMIP6 ensemble – surface temperature north of 50°N:**
Simulated differences in evapotranspiration in regions (land and ocean) north of 50°N and the respective CMIP6 ensemble spread. The black line shows the differences between WET and the DRY ($\Delta^{ts;+50N}_{|DRY-WET|}$), the red area gives the interquartile range ($IQR^{ts;+50N}$) — that is the difference between the 75ᵗʰ and the 25ᵗʰ percentile — of the CMIP6 ensemble, while the grey area provides 2 × the ensemble standard deviation ($\pm\sigma^{ts;+50N}$).

- I appreciate the effort of the authors to also conduct numerical experiments with ICON-ESM to substantiate their findings. However, I am not convinced that the way this is currently presented in the article is helpful for the readers. The ICON-ESM simulations are described in a dedicated Methods section 2.4, but not mentioned thereafter, neither in the results, nor in the discussion or conclusions. If it is a specific objective of the authors to substantiate their findings also with the simulations of ICON-ESM, then these should also be mentioned in the Results and Discussion sections. Otherwise, the authors might consider to omit presenting the ICON-ESM simulations at all in the present study.

For the sake of completeness we would still very much like to include the simulations with the ICON model. However, we agree that it may not have been ideal to present the simulations in the methods section but then not mention them throughout the entire manuscript. We think that it may be a solution to move everything concerning the ICON-based simulations to the supplementary materials and merely mention them at the very end of the main manuscript (line 707 of the revised manuscript):

*"Finally, please note that the present investigation relies exclusively on simulations with MPI-ESM. However, to confirm that our findings do not merely describe a specific feature of this particular model, but provide more general insights, we conducted an additional set of simulations with the new ICON-Earth System Model (ICON-ESM; Jungclaus et al., 2022). A brief overview of the respective results is included in the supplementary materials (Sec. S4)."*

- The final section which is labelled "Conclusions" provides mostly a discussion of the results, and in addition introduces further model results obtained using overshoot scenarios. However, the section does not present any concrete conclusions of the present work in a comprehensive and condensed way. I strongly suggest to carefully revise the structure and contents presented in this section. Connecting to my comment regarding the 1.5° simulations above, I suggest that these results should be presented in a dedicated subsection of the "Results" section. All results (land–atmosphere interactions, comparison to the CMIP6 spread, (optionally the ICON-ESM comparison), and the comparison of the overshoot runs) could then be discussed in a dedicated Discussion section building on the current "Conclusions" section. An additional section which provides the study's conclusions in a condensed way would further improve the article's accessibility and impact.

As stated above, we restructured the manuscript as suggested and included the stabilization simulations in the result section. Furthermore, we agree that our Conclusions read more like a Discussion and we have adapted the headings accordingly.

- The Discussion of the results does not sufficiently consider the parameterization of soil hydrology in other land surface schemes (see first comment above). An additional paragraph in which the present work with MPI-ESM is related to other model configurations could help to substantiate the findings regarding the inter-model spread also qualitatively.

Here, we included a very brief discussion of the problems with relating our model adaptations to the parameterization of soil hydrology in other land surface schemes, i.e. that it is really not clear what exactly is causing the large inter-model differences (line 655 of the revised manuscript):

*"As described in the methods section, our JSBACH setups were designed to reproduce "typical" inter-model differences and they manage to quantitatively capture the spread in the ensemble of PCN-MIP simulations reasonably well. However, it is exceedingly difficult to assess if our setups also adequately reflect the variability in the parametrizations employed by current-genertion land surface models, as it is highly uncertain what causes the diverging hydrological fluxes and states in the PCN-MIP participants. Andresen et al. (2020) suggested the representation of evapotranspiration, soil organic matter, the*

*water table – which in some models is used to estimate infiltration rates – and the vertical movement of water through the ground to be important sources of uncertainty in the models' soil hydrology schemes. However, even their detailed analysis did not resolve how specific implementations affect the hydrological processes or quantify the contribution of individual factors to the overall uncertainty.*

*One reason why it is so difficult to identify the key drivers of uncertainty is that it is often not one specific parametrization but the interactions between processes that determine the behaviour of a given model. For example, the impact of supercooling on the state of the land surface and the hydrological fluxes may depend strongly on the assumptions with respect to the mobility of the supercooled water and whether or not the water is assumed to be available to plants. This is particularly problematic as differences in the simulated soil hydrology may also originate from differences in the treatment of the soil thermal dynamics – as these determine the state of water in the soil – as well as from differences in the general model setup, e.g. vertical resolution and depth of the soil column, and the representation of vegetation. Thus, our results can merely estimate the bulk effect of the uncertainty included in the range of established soil hydrology representations without trying to connect them to specific formulations employed by present-day land surface models."*

**Specific Comments**

- l.109ff: The authors state that the model representation of soil physics is largely based on the modifications by Ekici et al. and explain the differences. However, it was not clear to me how the model version used relates to that described in de Vrese et al. (2021) where several modifications to the soil physics and biogeochemistry have been described. Maybe I overlooked this, but are these modifications also included in the model version used for the present study?

With respect to the soil physics, the model described in de Vrese et al. (2021) is essentially what we would consider the "most plausible setup" / "best-estimate parameter setting" of the model version used in the present study. In this model version, many of the optional formulations, which allowed us to create the WET and DRY simulations, were actually already included, but with the parameters chosen in such a way as to create a setup whose behaviour falls between the two extremes. At the same time the 2021-model version used a soil carbon model which vertically resolves the soil organic matter content, allowing us to consider the respective effects on the soil properties. This functionality is not included in the present version as it introduces so much additional uncertainty that it became unfeasible to use it in the fully coupled model, when performing ensemble simulations is extremely expensive. Thus, with regards to content and functionality, the present model version would have to be considered the direct successor of the Ekici et al. (2014) model and a predecessor of the 2021 model, hence, we describe the present model relative to the version of Ekici et al. (2014). We do apologize that this is somewhat confusing, but we think that describing our WET and DRY setups relative to the 2021-model would have been even less straight-forward.

*- l.124f: "but includes their influence on the hydrological soil properties." Does this refer to the modifications described in sections 2.1.2.-2.1.4 or are there further ways in which the organic matter influences the hydrological soil properties (e.g. hydraulic conductivities). If the latter is the case, all equations should be provided in the main text or the appendix. Otherwise it should be made clear, that this statement refers to the modifications described later on.*

Strictly speaking, all below-ground processes depend on the soil properties and are therefore affected by the explicit representation of soil organic matter. The processes described in section 2.1.3 are merely the ones, that were treated in a special way. This was clarified in the following way (line 122 of the revised manuscript):

*"Furthermore, our version does not limit the effects of the organic layer to the thermal processes but includes their influence on the hydrological states and fluxes. For most soil processes, this was done by assuming the properties of soil organic matter in the uppermost soil layer, with the parametrization of evaporation and transpiration being a notable exception (see Sec. 2.1.3)."*

However, we are not sure that including the equations for all processes affected by the presence of soil organic matter would improve the clarity of the manuscript as this would essentially mean to reproduce the largest part of the model documentation.

*- Section 2.1.1. mentions a lot of issues and problems which faced by soil hydrology parameterizations, in particular in JSBACH. However, it is not always clear which of these issues are being addressed in the present study and which are not.*

The main problem is the inability of many land surface models – including JSBACH – to adequately represent the effect of organic matter on evaporation and transpiration rates. Here, we actually failed to address one of the major shortcommings of our model, namely a missing representation of peatlands. We apologize for this omission and included the following in the revised manuscript (line 139 of the revised manuscript):

*"Additionally, JSBACH does not explicitly account for peatlands, which are the most common types of wetlands in the high northern latitudes, covering large areas especially in the West Siberian Lowlands and the region around the Hudson Bay (Olefeldt et al., 2021). In contrast to fresh litter, peat soils may feature very little connected pore spaces, which inhibits lateral drainage due to a low hydraulic conductivity. This detention of water results in shallow water tables in peatlands (Morris et al., 2022), with a high saturation of the near surface layer often leading to fully waterlogged soils and inundated surfaces."*

The present JSBACH setup can not address the issues described in section 2.1.1, in the sense that our approach could resolve any of the specific problems. We merely tried to be able to cover the range of potential impacts that a simplistic representation of organic matter may have on evapotranspiration. This we tried to clarify by modifying the final paragraph of section 2.1.1

(line 158 of the revised manuscript):

*"Without being able to represent the entirety of effects related to soil texture, including the low lateral drainage in peatlands, or to explicitly resolve the organic matter distribution horizontally and vertically, our model cannot adequately describe the impact of high soil organic matter concentrations on evaporation and transpiration rates. Instead, we include several options to capture the respective uncertainty, ranging from organic matter strongly inhibiting evaporation and transpiration to having only a minor effect on the respective fluxes (see below Sec. 2.1.3)."*

- l.158: It is not clear how infiltration, especially into frozen ground, is controlled in the current version. The authors only state that it is "exclusively controlled by the saturation of the near-surface soil layers ...", but they do not provide any equations or references to other work where this is described.

A brief description, including references, actually comes in the following sentence. Unfortunately, we started the latter with "Furthermore" which gave the misslieading impression that the the following was a separate point. We hope to have corrected this by changing "Furthermore" to "Here" (line 168 of the revised manuscript):

*"In the present version, this condition was removed so that infiltration is exclusively controlled by the saturation of the near-surface soil layers and the topography within the grid cell. **Here**, the ARNO model (Dümenil and Todini, 1992; Todini, 1996) that JSBACH uses to determine the infiltration rates based on the subgrid-scale orography, does not account for ponding effects. Instead, all water reaching the surface is either infiltrated or converted into surface runoff. In the present version of JSBACH, we implemented the WEtland-Extent-Dynamics (WEED) scheme, which adds an intermediate storage to the land surface that intercepts all rainfall and snow melt prior to infiltration or runoff generation (Stacke and Hagemann, 2012; de Vrese et al., 2021)".*

- l.225: The authors state that the field capacity constitutes the upper limit of the soil watercontent in JSBACH, which I found confusing. I would have expected that it is possible to have soil water contents exceeding the field capacity, for example under saturated conditions. For example, in l.248 it is written that "only the water that exceeds the layer's field capacity is added to the drainge flux" which in my view contradicts the formulation in l.225. Am I missing something here? A clarification would be appreciated.

It usually takes a couple of days for excess water to drain from the soil. Thus, Dr. Nitzbon is correct that, in reality, the water content of the soil frequently exceeds the field capacity. In JSBACH however the excess water is removed at the end of each time step and added to the drainage flux. This explicit removal of water however rarely occurs in the standard version of the model as the vertical transport close to saturation in combination with the lateral drainage is usually very efficient at moving water through and out of the soil, including the infiltration peaks following rain events. However, in the present model version,

with the lateral drainage severely curtailed by an ice impedance factor, water frequently accumulates at the bottom of the active layer and the water exceeds the field capacity at the end of the time step. What may have substantially added to the confusion, is that we actually referred to the wrong ice impedance factor in line 248, instead of $F_{perc,l}$ it should have been $F_{drain,l}$. We apologize for this mishap.

- l.292: It is not clear to which version „JSBACH's original formulation" refers. It is stated that the original formulation would prohibit infiltration at sub-zero temperatures, but I understood that this had been only introduced into the model by Ekici et al. Also in light of my comment below on using the term "reference model", it might be helpful to clearly designate the different JSBAHC model versions/configurations referred to in this paper.

Here, we apologize for the confusion. Unfortunately, the development of JSBACH did and does not always follow a linear trajectory which makes the nomenclature somewhat difficult. Especially, the fact that the standard model version (the CMIP6 model) is several years younger than the implementation of Ekici et al. (2014), but does not feature any of the latter's functionality is somewhat counterintuative. To make matters slightly less confusing we now clarify that the standard version indeed referes to the CMIP6 model (line 112 of the revised manuscript):

*"Please note that throughout the manuscript, "standard model" does not refer to the implementation of Ekici et al. (2014) but to the CMIP6 version of the model described in Reick et al. (2021)."*

With respect to the issue of infiltration: The temperature-condition has been a part of JSBACH for a long time, while Ekici et al. (2014) introduced a new snow scheme. The latter led to cooler soil temperatures during spring, hence the unexpected interaction with the temperature-condition.

- l.314ff: It is not clear how the subsurface parameters have been set, since the deep model configuration deviates from the standard one. I suspect that the same datasets as in González-Rouco et al., 2021 and Steinert et al. 2021a have been used? This should be clarified in order to allow reproducibility.

In contrast to some of the experiments performed in the above-mentioned studies, we actually did not deviate from the standard setup regarding the depth of the hydrologically active parts of the soil. This we now clarify in the methods section (line 350 of the revised manuscript):

*"Note that the downward extension of the soil column has no direct impact on the subsurface hydrology as we assume the ground to consist of impermeable bedrock below the soil depth of the standard model configuration."*

- l.414: Why not stating the proportion of simulations contained in two standard deviations ( 95%) if these are used?

Again we have to apologize if this formulation was a bit misleading. Our definition of the spread is not two standard deviations of the mean – which is indeed 95% given a normal distribution – but rather two standard deviations in total (one above and one below the mean). The latter encompasses roughly two thirds of the ensemble as stated in the methods section. We tried to clarify this (line 418 of the revised manuscript):

*" ... as well as the range contained within one ensemble standard deviation of the mean ($\pm\sigma$) — covering about two thirds of the ensemble, if the latter is normal distributed."*

More importantly, we changed the respective abbreviation from "$2\sigma$" to "$\pm\sigma$" throughout the manuscript.

- l.422/Fig. 5: Here and later the authors often refer to "trends" which are shown in Fig. 5. How have these trends been estimated? (linear regression?, which time period?, ...). Please specify this, ideally already in the Methods section.

In the captions of figure 5 and 9 (formerly Fig. 8) we clarified that:

*"Trends were estimated applying a simple linear regression to the values covering the period 2000 - 2099."*

- l.433ff: I do not understand the argumentation regarding the effect of excess ice. I would expect that, in a first-order approximation, the inclusion of excess ice and associated ground subsidence would result in wetter conditions if thaw depths increase, since the ground subsidence counteracts the possibility of deeper infiltration (see Fig. 8 in Andresen et al. (2020)).

This does depend a lot on how "wetter conditions" are defined. Indeed subsidence may counteract the desiccation of the soil layers at the surface, leading to more saturated conditions at the top of the soil column. However, the total amount of water contained in the vertical column is lower because the depth of the hydrologically active soil / soil above the bedrock is shrinking. This is also visible in Fig. 8 in Andresen et al. (2020), where the sum of soil water and soil ice, decreases during the thaw / subsidence process.

- The final paragraph of the results section (l.538ff) introduces an additional simulation setup (W2D). Admittedly, I had a hard time understanding this paragraph. This part would benefit from a bit more detailed explanations of the motivation for and the interpretation of the additional simulations. It might also help to replace some of the rather technical language which uses symbols as parts of the sentences by verbal descriptions.

We moved the more detailed description of the W2D setup to the methods section and removed the discussion of the effects north of 50°N. Furthermore, we tried to provide a more verbal description of our findings, hoping that this makes the paragraph more readily accessible (line 554 of the revised manuscript).

*"This raises the question whether the uncertainty in the permafrost hydrology is insufficient to explain the increase in the inter-model spread, especially during the second half of the century, or whether the issue of constant temperature differences is specific to our setups. Per design, the simulations with the WET and the DRY setup become more similar over time as many of the distinctions depend on the soil ice content, which decreases throughout the simulation. To test how far the lack of trends is related to this design-feature, we compared the WET simulation to a simulation with the W2D setup. In the W2D simulation the parametrizations switch from WET to DRY when the near surface permafrost disappears in a given grid cell, with the W2D simulation becoming increasingly different from the WET simulation. With respect to precipitation and surface temperature, the differences between W2D and WET exhibit trends that are substantially larger than those in the differences between DRY and WET (Fig. 9b,c). The trend in $\Delta^{pr}_{|W2D-WET|}$ is also larger than the trend in and $IQR^{pr}$, closely matching the trend in $\pm\sigma^{pr}$, while the trend in the temperature differences, $\Delta^{ts}_{|W2D-WET|}$, is very similar to the trend in $IQR^{ts}$. This indicates that differences in soil hydrology parametrizations may, in principle, contribute to the trend of the inter-model spread, given that the parametrizations do not become more similar with the advancing permafrost degradation. Here, however, the trends in $\Delta^{pr}_{|W2D-WET|}$ and $\Delta^{ts}_{|W2D-WET|}$ stem from a much larger causative trend in the evapotranspiration differences, $\Delta^{evap}_{|W2D-WET|}$ (Fig. 9a). The latter is almost twice as large as the trend in $\pm\sigma^{evap}$ and almost 20 times larger than the trend in $IQR^{evap}$, strongly suggesting that the trends in the temperature-spread of the CMPI6 ensemble are not caused exclusively by the divergence of evapotranspiration rates (Hahn et al., 2021)."*

The description of the W2D setup in the method section now reads (line 325 of the revised manuscript):

*"Although the manuscript focuses on the two setups described above, we performed a third, highly synthetic setup — the W2D setup – which exhibits increasingly dry conditions under a future warming. This setup assumes that the characteristics of the soil hydrology are determined by the presence of near-surface permafrost and change when the latter is degraded. All grid cells start with the parametrizations of the WET setup and the configuration is maintained as long as the model simulates permafrost in the upper 3 m of the soil. However, the parametrizations switch from WET to DRY (with the exception of the soil depths and the maximum wetland retention $P_{lag}$) whenever the annual maximum thaw depth in the grid-cell extends beyond a depth of 3 m. For the high-emission scenario considered in this study, the majority of the grid cells in the northern permafrost regions transitions from WET to DRY during the 21$^{st}$ century, with the W2D simulation becoming increasingly different from the WET simulation."*

- l.589: It might be interesting to explicitly state the high latitudes become warmer under the DRY relative to the WET setup while the tropics and mid latitudes are colder under the WET setup relative to the dry setup.

Here, we included the following (line 603 of the revised manuscript):

*"And because the albedo-differences affect primarily the Arctic and subarctic region, this global mean temperature is reached with a substantially different latitudinal distribution – with the DRY simulation exhibiting predominantly higher temperatures in the northern high- and mid latitudes, while the temperatures are significantly lower throughout tropics (Fig. 12a)."*

- l.603f: Please provide references for the statement regarding the effect of sea ice.

Here, we included the following references (line 620 of the revised manuscript):

For a given model, and for the observational record, a clear, linear relationship between global mean temperature and Arctic sea-ice coverage has long been identified across all months (Gregory et al., 2002; Mahlstein and Knutti, 2012; Niederdrenk and Notz, 2018).

- l.615: It should be stated in the text that the subsea permafrost was diagnosed using a different model configuration.

Here, we included the following (line 632 of the revised manuscript):

*"Here, the subsea permafrost extent near the sea bottom is strongly affected by the temperature differences between DRY and WET, in particular in the Laptev Sea and the East Siberian Sea where the frozen fraction in the top 10 m of the subsurface differs by up to 50 % (Fig. 12d, note that the subsea permafrost was diagnosed with a different model version, which is described in Wilkenskjeld et al. (2021))."*

- l.624: It is not clear what "negatively affected" means in this context.

Here, we clarified that (line 642 of the revised manuscript):

*"However, not all tipping elements are closer to the critical threshold due to the smaller temperature gradient in DRY."*

- l.662: The work of Aas et al. (2019) might also be relevant in this context.

Here, we included a reference to the manuscript by Aas et al. (2019), along with the references to Cresto Aleina et al. (2013); Nitzbon et al. (2021); Smith et al. (2022).

- l.664f: It would be interesting to discuss whether ICON-L/JSBACH4 provide this functionality.

In principal, JSBACH4 is extremely well suited to accommodate such a tiling approach. However, at the moment the structure of the model is set up to merely represent plant functional types – similar to the functionality of JSBACH3. Whether or not a multilayered tiling will be developed for the model is highly uncertain as this is largely depending on strategical decisions taken at the developing institutes. Therefore, we would prefer not to discuss the potential of ICON-L.

- To my opinion, Figure 3 is not crucial for understanding the methodology and should go into the appendix or supplement.

We agree with Dr. Nitzbon and have replaced the figure with the schematic depiction of the WET and DRY setup (see Fig. 3).

- The caption of Fig. 4 refers to a "reference model" abbreviated REF, which is, however, not introduced at any point in the main text. It is not clear whether this is the standard configuration of MPI-ESM/JSBACH3.2 or whether it includes the modifications by Ekici et al. Please clarify.

This indeed refers to the standard model, i.e. JSBACH3.2, which we have clarified in the figure caption:

*"Shown is the partitioning for the standard model (REF, top), the DRY (left) and the WET setup (right)."*

- Fig. 5/6: While Fig. 5 shows the trends in total water content, Fig. 6i shows the aggregated changes in liquid water content. However, it would also be interesting to see how the absolute values of total water content evolve during the 21st century and how the two setups compare to each other to get the full picture.

The development of the total soil water content shows that, while there is a negative trend in WET and a positive trend in DRY, the total soil water in WET remains larger than in DRY throughout the 21st century (see Fig 999 below). However, we do not think that this information is critical for our findings and we would prefer not to include the additional figure in the manuscript.

- Fig. 6: The differences in permafrost volume (panel h) are quite huge among the two setups. It would be interesting to also show the simulated permafrost areal coverage and mention how it compares to estimates for the present day coverage. Are other estimates contained within the spread between the WET and DRY scenarios?

[Figure]

**Figure 999. Simulated total soil water content during the 21st century.**
Note that the figure is not included in the revised manuscript.

We agree with Dr. Nitzbon that information on the areal extent of the near surface permafrost may be interesting, as this is the variable most often compared between different models. However, we would prefer to not include an additional figure but merely report the respective numbers in the text (line 482 of the revised manuscript):

*"(Fig. 6h; Note that the near surface permafrost volume corresponds to an initial permafrost area of about $22\times10^6$ km² in WET and about $16\times10^6$ km² in DRY. At the end of the 21st century the values are $11\times10^6$ km² and $4\times10^6$ km² respectively)"*

- Fig. 7: I do not understand why this figures uses a mix of "area" and "line" plot features. It would make more sense if the IQR and 2sigma would be indicated as areas, and the Delta as lines. But showing all quantities as a line plot should also work.

Many aspects of this figure were decided based on personal preferences. And as there was no scientific reason for showing the Deltas as a colored area, we will follow Dr. Nitzbon's suggestion and show IQR and 2sigma as (colored and grey) areas, and the Delta as a line.

- Fig. 7/8: To be consistent and to provide the full picture, you might consider to include the plots and values for the region north of 50°N also for precipitation and evapotranspiration.

As we discuss the remote effects – that is all effects outside of the terrestrial, non-glaciated permafrost affected areas – as a result of the shift in latitudinal temperature gradient, we would prefer to not include a discussion of the effects on evapotranspiration and precipitation in the region north of 50°N.

- Fig. 9: The scenario (SSP5-8.5) should be mentioned in the caption.

In the figure caption, we clarified that the graph shows:

*"Simulated global mean surface temperatures for the DRY (red), the WET (blue) setup and range of the CMIP6 ensemble mean $\pm$ one standard deviation for the SSP5-8.5 scenario."*

- Tab. 2 should go into the supplement.

The table was moved to the supplementary materials.

**Technical corrections**

- l.29: Should be "as contained in the Earth's atmosphere."

The text was modified accordingly.

- l.137: "be" missing in "could derived from observations".

The text was modified accordingly.

- l.150: "see below" –> Please be more specific and refer to specific sections.

Here, we clarified that (line 160 of the revised manuscript):

*Instead, we include several options to capture the respective uncertainty, ranging from organic matter strongly inhibiting evaporation and transpiration to having only a minor effect on the respective fluxes (see below Sec. 2.1.3)."*

- l.157: typo: "soil" instead of "seoil"

The text was modified accordingly.

- Fig. 5: I don't see why you write "EVAP.", "PR." and "H20" instead of "Evaporation", "Precipitation" and "Total soil water" in the heading.

Using the abbreviations allowed us to use a large font-size while maintaining a comparatively compact layout. However, we are happy to spell out the terms if it makes the figure more accessible.

[Figure]

**Figure 5. Arctic futures**

[revised manuscript text omitted]

---

## Author Comment (AC2)

**Response to comments of reviewer 2**

Before replying to the individual comments, we would like to sincerely thank the reviewer for the time and effort she/he has taken to review our submission. We hope that we managed to make good use of all suggestions and think that especially the (brief) discussion of the seasonality of effects has improved the quality of our manuscript.

*Please note that, in the following point by point address, we repeat the reviewer's comments in red letters while **our response** is given in black letters. All the post-reply updates are indicated in blue.*

The study "Representation of soil hydrology in permafrost regions may explain large part of inter-model spread in simulated Arctic and subarctic climate" by de Vrese et al. evaluates the impact of model settings influencing the hydrology in boreal and Arctic regions in the land surface model JSBACH. The purpose of the study is mainly to showcase the significant consequences of these settings on the simulated regional and even global climate, not so much to present new ways to resolve these shortcomings. But that is to be expected, given the magnitude of the problem.

The study follows a logic setup and the results are of great interest to the readers of TC. The manuscript is well written and I recommend publication after addressing the following points:

**Major comment:**

The authors focus the results on annual averages, but in most cases do not present the seasonality, which is important for understanding the changes. Two concrete examples in the following, but this aspect should be taken into account more throughout the entire manuscript:

In general, we would prefer to stick to showing and discussing the annual means as much as possible, as we think that the main findings and the general message of the manuscript do not require the additional detail. However, there are two important exceptions, where we agree with the reviewer that details with respect to the seasonal variability are indeed crucial. On one hand, low clouds affect the surface energy balance differently during summer than during winter. Thus, it is important to point out that the differences in the soil hydrology have an effect on the cloud cover mainly during spring and summer. On the other hand, the seasonal variability of the temperature differences clearly shows that the differences in soil hydrology can not explain

the large ensemble spread during the snow covered period. Here, we think that these to points can be made using the examples provided by the reviewer.

L. 449 ff: This is a really important paragraph which should be extended by presenting the seasonality of the effects. As an example, 0.2mm per day corresponds to about 70mm per year, but I guess this difference will mostly accumulate in summer and fall when most of the evaporation occurs? Are there differences in the snowfall which could possibly affect the insulation of the snowpack and thus ground temperatures? Same for the cloud feeback, total incoming radiation (short- plus long-wave) should generally reduce for cloudy skies in summer, but increase in winter. So when is cloudiness increased, all year or mainly in summer?

While there are differences in the simulated snowfall and -cover, the most important factor is the seasonality of the differences in cloudiness. The cloud radiative effect is very different between the summer and winter cloud cover, hence, we agree that it is important to mention that the additional evapotranspiration in WET leads to an increased cloudiness almost exclusively during spring and summer (line 460 of the revised manuscript):

*Furthermore, the differences in relative humidity result in differences in the cloud cover which constitute another important feedback on the surface energy balance (Fig. 6e). The increased cloudiness in WET occurs mainly during the snow free period – spring to early fall in the southern permafrost regions, with the length of the period decreasing in northward direction – when the surface reflectivity is determined by a comparatively dark vegetation cover and similarly dark bare soil areas. Thus, the more extensive cloud cover notably raises the planetary albedo (relative to DRY), reducing the surface incoming solar radiation by between $10\,W\,m^{-2}$ at the beginning and $13\,W\,m^{-2}$ at the end of the $21^{st}$ century.*

L. 519 ff: same here, would be really nice to present the seasonality of the effects. The authors emphasize the importance of permafrost many times, and the winter aspect, especially the snow cover, is highly important for permafrost occurrence and thaw.

We now include Fig. 8 (see below), showing the seasonality of the WET-DRY differences and the CMIP6 ensemble spread. For evapotranspiration and precipitation the WET-DRY differences and the ensemble spread are reasonably well correlated. However this is not the case for the surface temperatures, indicating that there are additional factors – most likely differnces in the parametrization of the snow cover – determining the peak in the ensemble spread. This we now discuss together with the results based on the annual mean values (line 508 of the revised manuscript):

*In the case of precipitation, the differences in the soil-hydrology parametrizations appear to offer a large explanatory potential (Fig. 7b). On average $\Delta^{pr}_{|DRY-WET|}$ amounts to about $0.16\,mm\,day^{-1}$, $IQR^{pr}$ to about $0.19\,mm\,day^{-1}$ and $\pm\sigma^{pr}$ to $0.32\,mm\,day^{-1}$. Thus, even if considering $\pm\sigma^{pr}$ to be the more appropriate measure, about half of of the inter-model spread of*

[Figure]

**Figure 8. Comparison to CMIP6 ensemble – seasonality:**

a) Simulated differences in evapotranspiration in permafrost regions and the respective CMIP6 ensemble spread. The black line shows the differences between WET and the DRY ($\Delta^{evap}_{|DRY-WET|}$), the green area gives the interquartile range (IQR$^{evap}$) — that is the difference between the 75$^{th}$ and the 25$^{th}$ percentile — of the CMIP6 ensemble, while the grey area provides $2 \times$ the ensemble standard deviation ($\pm\sigma^{evap}$). b) same as a but for precipitation ($\Delta^{pr}_{|DRY-WET|}$, IQR$^{pr}$, $\pm\sigma^{pr}$), c) surface temperatures in permafrost grid cells ($\Delta^{ts}_{|DRY-WET|}$, IQR$^{ts}$, $\pm\sigma^{ts}$). Shown is the seasonality averaged over the 21$^{st}$ century.

*the CMIP6 ensemble may be explainable by diverging evapotranspiration rates resulting from differences in the parametrizations of the permafrost hydrology. Here, $\Delta^{pr}_{|DRY-WET|}$ exhibits a marked peak in the summer months, when the causative differences in evapotranspiration are largest $\Delta^{evap}_{|DRY-WET|}$ (Fig. 8a,b). A similar behaviour can be seen for the ensemble spread, even though the (relative) seasonal variations are less pronounced, especially in case of IQR$^{pr}$. With regards to the surface temperatures, $\Delta^{ts}_{|DRY-WET|}$ matches the overall magnitude of the CMIP6 ensemble spread similarly well (Fig. 7c), with $\Delta^{ts}_{|DRY-WET|}$ being equal to IQR$^{ts}$ (2.6 °C) and representing about two thirds of $\pm\sigma^{ts}$ (3.7 °C). However, as with the differences in precipitation, $\Delta^{ts}_{|DRY-WET|}$ peaks – at 4.2 °C – when the differences in evapotranspiration are largest (Fig. 8c), which is not the case for IQR$^{ts}$ and $\pm\sigma^{ts}$. While the latter exhibit a notable increase in summer, their annual maximum occurs during winter – with 3.8 °C and 7.2 °C respectively – when $\Delta^{ts}_{|DRY-WET|}$ is the lowest. As described above (Sec. 3.1), the differences between WET and DRY mainly originate from a divergence of the cloud cover and the resulting differences in the planetary albedo. But as there are only minor differences in cloudiness – and very little solar radiation – during the snow cover season, this feedback is not present during the winter months. More importantly, the cloud radiative effect differs between the snow covered and the snow free period. The albedo of clouds is similar to those of snow and ice covered surfaces and an increase in cloudiness does not lower the planetary albedo in winter. Instead, (low) clouds are more likely to increase the surface temperatures as they raise the surface net radiation by reflecting the longwave radiation emitted by the surface (Vihma et al., 2016). Thus, it is mainly the differences in soil heat content – resulting from differences in the energy uptake*

*during the snow free period – and differences in latitudinal heat transport (not shown) that sustain $\Delta^{ts}_{|DRY-WET|}$ during winter, while it is most likely differences in the parametrization of the snow and ice albedo determining the large ensemble-spread (Menard et al., 2021). Consequently, the explanatory power of the differences in the soil-hydrology parametrizations appears to be limited to the snow free period.*

**Minor comments:**

-Introduction: there are very relevant model studies on paleoclimate that should be cited, e.g. Renssen, H., Isarin, R. F. B., Vandenberghe, J., Lautenschlager, M., & Schlese, U. (2000). Permafrost as a critical factor in paleoclimate modelling: the Younger Dryas case in Europe. Earth and Planetary Science Letters, 176(1), 1-5.

It is true that the representation of the permafrost related physics has most certainly had an impact on our ability to simulate the climate since LGM. However, as our investigation focusus exclusively on present-day conditions and the scenario period, we would prefer not to make the digression into the field of paleoclimatic modelling.

-Sect. 2.2 Why is the W2D setup not presented here?

We agree that the methods section is the better place to introduce the W2D setup and moved the description accordingly (line 325 of the revised manuscript):

*"Although the manuscript focuses on the two setups described above, we performed a third, highly synthetic setup — the W2D setup – which exhibits increasingly dry conditions under a future warming. This setup assumes that the characteristics of the soil hydrology are determined by the presence of near-surface permafrost and change when the latter is degraded. All grid cells start with the parametrizations of the WET setup and the configuration is maintained as long as the model simulates permafrost in the upper 3 m of the soil. However, the parametrizations switch from WET to DRY (with the exception of the soil depths and the maximum wetland retention $P_{lag}$) whenever the annual maximum thaw depth in the grid-cell extends beyond a depth of 3 m. For the high-emission scenario considered in this study, the majority of the grid cells in the northern permafrost regions transitions from WET to DRY during the 21$^{st}$ century, with the W2D simulation becoming increasingly different from the WET simulation."*

-L. 251 ff, "supercooled water": this implementation seems weird, the increasing ice content should simply decrease the hydraulic conductivity, thus limiting and finally suppressing both vertical and lateral flow? I agree that setting water mobility to zero for temperatures below the freezing point is better than allowing free flow as in unfrozen soils, but making the hydraulic conductivity dependent on water/ice content should be really straight-forward solution.

Indeed this is also what we expected when we build the model. However, we had to find out that in some regions a power-law-based ice impedance factor alone may be insufficient. We do apply such a factor to percolation and lateral drainage in the WET simulation (in DRY only to percolation). However, in clay rich soils a large fraction of the water may remain liquid when temperatures are below but close to the melting point. In this constellation an ice impedance factor substantially reduces but does not completely stop all movement of water. Consequently – though a very slow process – the soils can loose up to 20% of the soil water during winter. Thus, obtaining highly saturated soils with a perched water table with our model required to immobilize the supercooled water, which was actually done by including the supercooled water (essentially as extra ice) in the calculation of the ice-impedance factor (line 271 of the revised manuscript).

*"In the implementation of Ekici et al. the supercooled water behaves similarly to water at temperatures above 0°C — that is, it can percolate through and drain from the soil. In the present version, the movement of supercooled water can be limited in the presence of ice, using the above described ice-impedance factors. However, some water movement, is still possible and especially clay-rich soils can loose a large fraction of the supercooled water over longer periods. To prevent this cold season drying of the soils from happening, we additionally included the option to "immobilize" the supercooled water, essentially prohibiting all fluxes below the freezing point."*

-Fig. 8: My opinion, but I think the figure would be easier to read if the authors selected a more "traditional" design. Since the graphs are labelled a,b,c,d anyway, they could for example distinguish the individual bars in each plot by colors, but re-use the same colors in all plots.

Here, we (partly) followed the reviewers advice and re-use the same colors in the individual subplots (Fig. 9). However, as we would prefer to retain the same color-scheme as in Fig. 7 and 8, we show the bars of IQR in the color of the respective variables, i.e. green for evapotranspiration, blue for precipitation and yellow for surface temperatures. Please also note that, following the suggestions of reviewer 1, we restructured the manuscript and introduced a dedicated results section for the impacts outside of the terrestrial permafrost regions. Consequently, the differences in the region north of 50°N are no longer included in this figure – i.e. we removed panel d.

-Fig. 10 i: explain the grey area in the caption

In the figure caption (to now figure 12), we clarify that:

*"In panel a - f and h, areas where differences are not significant (p-value > 0.05) are hatched, while subfigure g shows the minimum 3-year mean precipitation over land during the 50 year period without test of significance. Dark grey areas in panel i show the bathymetry of the Atlantic ocean."*

[Figure]

**Figure 9. Comparison to CMIP6 ensemble – 21$^{st}$-century trends:**

a) Trends in evapotranspiration during the 21$^{st}$ century: $2 \times$ the CMIP6 ensemble standard deviation, the interquartile range of the CMIP6 ensemble, differences between WET and DRY and differences between W2D and WET — averaged over the northern permafrost regions. b) Same as a) but for precipitation and c) surface temperatures.

**References**

Menard, C. B., Essery, R., Krinner, G., Arduini, G., Bartlett, P., Boone, A., Brutel-Vuilmet, C., Burke, E., Cuntz, M., Dai, Y., Decharme, B., Dutra, E., Fang, X., Fierz, C., Gusev, Y., Hagemann, S., Haverd, V., Kim, H., Lafaysse, M., Marke, T., Nasonova, O., Nitta, T., Niwano, M., Pomeroy, J., Schädler, G., Semenov, V. A., Smirnova, T., Strasser, U., Swenson, S., Turkov, D., Wever, N., and Yuan, H.: Scientific and Human Errors in a Snow Model Intercomparison, Bulletin of the American Meteorological Society, 102, E61–E79, https://doi.org/10.1175/bams-d-19-0329.1, 2021.

Vihma, T., Screen, J., Tjernström, M., Newton, B., Zhang, X., Popova, V., Deser, C., Holland, M., and Prowse, T.: The atmospheric role in the Arctic water cycle: A review on processes, past and future changes, and their impacts, J. Geophys. Res. Biogeosci., 121, 586–620, 2016.